# Addressing Misspecification in Simulation-based Inference through Data-driven Calibration

Antoine Wehenkel [* 1]    Juan L. Gamella [* 2 3]    Ozan Sener [1]    Jens Behrmann [1]    Guillermo Sapiro [1]
Jörn-Henrik Jacobsen [2]    Marco Cuturi [1]

## Abstract

Driven by steady progress in deep generative modeling, simulation-based inference (SBI) has emerged as the workhorse for inferring the parameters of stochastic simulators. However, recent work has demonstrated that model misspecification can compromise the reliability of SBI, preventing its adoption in important applications where only misspecified simulators are available. This work introduces robust posterior estimation (RoPE), a framework that overcomes model misspecification with a small real-world calibration set of ground-truth parameter measurements. We formalize the misspecification gap as the solution of an optimal transport (OT) problem between learned representations of real-world and simulated observations, allowing RoPE to learn a model of the misspecification without placing additional assumptions on its nature. RoPE demonstrates how OT and a calibration set provide a controllable balance between calibrated uncertainty and informative inference, even under severely misspecified simulators. Results on four synthetic tasks and two real-world problems with ground-truth labels demonstrate that RoPE outperforms baselines and consistently returns informative and calibrated credible intervals.

## 1 Introduction

Many fields of science and engineering have shifted in recent years from modeling real-world phenomena through a few equations to relying instead on highly complex computer simulations. While this shift has increased model versatility and the ability to explain or replicate complex phenomena, it has also necessitated the development of new statistical inference methods. In particular, state-of-the-art simulation-based inference (SBI, Cranmer et al., 2020) algorithms leverage neural networks to learn surrogate models of the likelihood (Papamakarios et al., 2019), likelihood ratio (Hermans et al., 2020), or posterior distribution (Papamakarios & Murray, 2016), from which one can extract confidence or credible intervals over the parameters of interest given an observation. While SBI has proven helpful when the simulator is a faithful description of the studied phenomenon, e.g., for scientific applications (Delaunoy et al., 2020; Brehmer, 2021; Lückmann, 2022; Linhart et al., 2022; Hashemi et al., 2022; Tolley et al., 2023; Avecilla et al., 2022), recent work has also highlighted the unreliability of SBI methods under model misspecification (Cannon et al., 2022; Schmitt et al., 2023).

**Addressing Misspecification with a Calibration Set.** In this work, we target important applications of SBI in common settings where (1) the goal is to estimate a hard-to-measure variable from indirect but readily available measurements of other variables; (2) only misspecified simulators relating these variables are available; and (3) a few ground-truth pairings of the hard-to-measure variables and the related variables are available in a *calibration* set[1]. Such a setting can arise, for example, when inferring the properties of a patient's cardiovascular system from non-invasive and abundant measurements of other physiological signals (Wehenkel et al., 2023), or when developing soft sensors to monitor industrial processes in real time, where directly measuring the quantity of interest is costly and time consuming, for example, through laboratory analysis, but where related variables can be measured quickly and inexpensively (Jiang et al., 2021; Perera et al., 2023).

**Our Contributions.** We introduce robust posterior estimation (RoPE), an algorithm that addresses model misspecification to provide accurate uncertainty quantification for the parameters of black-box simulators. In such misspecified

---

[*]Equal contribution   [1]Apple [2]Work done while being at Apple [3]ETH Zürich. Correspondence to: Antoine Wehenkel <awehenkel@apple.com>.

*Proceedings of the 42nd International Conference on Machine Learning*, Vancouver, Canada. PMLR 267, 2025. Copyright 2025 by the author(s).

[1]Note that our use of the term *calibration set* should not be confused with its usage in the context of model mis-calibration in well-specified SBI (Hermans et al., 2022), as we clarify in Appendix A.

settings, the main challenge lies in the absence of a paired dataset of simulated and corresponding real outputs. To handle this knowledge gap, RoPE estimates a coupling between real and simulated observations using optimal transport (OT, Peyré et al., 2017; Villani et al., 2009). The algorithm extends neural posterior estimation (Papamakarios & Murray, 2016) and models misspecification using OT. We evaluate the performance of the algorithm on existing benchmarks from the SBI literature and introduce four new benchmarks, two of which are synthetic and two come from real physical systems. To the best of our knowledge, the latter constitute the first real-world benchmarks that directly provide a ground truth for the inferred parameters for SBI under misspecification. We conduct additional experiments to investigate the impact on RoPE's performance of varying calibration set sizes, prior misspecification, and distribution shifts, as well as various ablation studies.

## 2 Background & Notation

In this section, we first pose the machine learning problem we are trying to solve and then formally introduce SBI, model misspecification, and OT, as our method lies at the intersection of these fields.

We consider a simulator, $S : \mathbb{R}^k \times [0, 1] \to \mathbb{R}^d$, that takes in physical parameters $\theta \in \Theta \subseteq \mathbb{R}^k$ and a random seed $\varepsilon \in [0, 1]$ to generate simulated measurements $\mathbf{x}_s \in \mathcal{X}_s \subseteq \mathbb{R}^d$. The simulator is a simplified version of a real and unknown generative process $p^\star(\mathbf{x}_o) := \int p^\star(\theta) p^\star(\mathbf{x}_o \mid \theta) \mathrm{d}\theta$ that produces real-world observations $\mathbf{x}_o \in \mathcal{X}_o \subseteq \mathbb{R}^d$. We assume this process depends on parameters with the same physical meaning as the ones of the simulator and thus use the same notation $\theta$. Our task is to estimate a well-calibrated and informative posterior distribution $p(\theta \mid \mathbf{x}_o^i)$ for each observation $\mathbf{x}_o^i$ in a test set $\mathcal{D}$, reducing uncertainty compared to the prior distribution, assuming that the prior is well-specified. To achieve our goal, we have access to: **1.** the misspecified simulator $S$ that embeds domain knowledge and generates samples whose distribution approximates $p^\star(\mathbf{x}_o \mid \theta)$, **2.** a prior $p(\theta)$ that approximates the marginal distribution $p^\star(\theta)$ of parameters in the real-world, **3.** a small calibration set of labeled real-world observations $\mathcal{C} := \{(\theta^i, \mathbf{x}_o^i)\}_{i=1}^{n_c}$ composed of i.i.d. samples from $p^\star(\theta, \mathbf{x}_o)$, which enables data-driven correction of the simulator's misspecification, **4.** a test set $\mathcal{D} := \{\mathbf{x}_o^i\}_{i=1}^{n_o}$ of real-world observations arising from $p^\star(\mathbf{x}_o)$ for which we want to estimate the posterior.

### 2.1 Simulation-based Inference (SBI)

Applying statistical inference to simulators is challenged by the absence of a tractable likelihood function (Cranmer et al., 2020). As a solution, SBI algorithms leverage modern machine learning methods to tackle inference in this likelihood-free setting (Lueckmann et al., 2021; Delaunoy et al., 2022; Glöckler et al., 2022). Among SBI algorithms,

neural posterior estimation NPE (Papamakarios & Murray, 2016; Lueckmann et al., 2017; Radev et al., 2020) is a broadly applicable method that trains a conditional density estimator of $p(\theta \mid \mathbf{x}_s)$ from a dataset of parameter-simulation pairs. In this paper, we focus on making NPE robust to model misspecification.

NPE usually parametrizes the posterior with a neural conditional density estimator (NCDE), which is composed of (1) a neural statistic estimator (NSE), denoted by $\mathbf{h}_\omega : \mathcal{X}_s \to \mathbb{R}^l$, that compresses observations into $l$-dimensional representations and, (2) a normalizing flow (NF, Papamakarios et al., 2021; Tabak & Vanden-Eijnden, 2010) that parameterizes the posterior density as $p_\phi(\theta \mid \mathbf{h}_\omega(\mathbf{x}_s))$. The parameters $\phi$ and $\omega$ of the NCDE are trained with stochastic gradient ascent on the expected log-posterior probability, solving the following optimization problem

$$\phi^\star, \omega^\star \in \arg\max_{\phi, \omega} \mathbb{E}_{\substack{\theta \sim p(\theta) \\ \varepsilon \sim \mathcal{U}[0,1]}} \left[ \log p_\phi(\theta \mid \mathbf{h}_\omega(S(\theta, \varepsilon))) \right], \quad (1)$$

where $p(\theta)$ denotes a prior over the parameters $\theta$.

Under the assumption that the class of functions represented by the NCDE contains the true posterior, solving (1) leads to a surrogate $p_{\phi^\star}(\theta \mid \mathbf{h}_{\omega^\star}(\mathbf{x}_s))$ that matches exactly the posterior $p(\theta \mid \mathbf{x}_s)$ corresponding to the simulator. In that case, $\theta \perp \mathbf{x}_s \mid \mathbf{h}_{\omega^\star}(\mathbf{x}_s)$, that is, the NSE $\mathbf{h}_{\omega^\star}$ is a sufficient statistic of $\mathbf{x}_s$ for the parameter $\theta$ (Chen et al., 2020; Wrede et al., 2022; Chan et al., 2018). In practice, we can only approach perfect training by generating a sufficiently large number of pairs $(\theta, \mathbf{x}_s)$ and doing a search on the NCDE's architecture and training hyperparameters. To simplify notation, we denote the NCDE learned with NPE as $\tilde{p}(\theta \mid \mathbf{x}_s)$.

### 2.2 Model Misspecification

In statistics, where the model parameters do not necessarily carry real-world meaning, model misspecification generally denotes the inability of a model to reproduce the observed data distribution. Formally, a parametric model $p(\mathbf{x}_o \mid \theta)$ is said to be misspecified with respect to some true data-generating process $p^\star(\mathbf{x}_o)$ if the latter does not fall within the family of distributions defined by the model, i.e. $\nexists \theta \in \Theta : p(\mathbf{x}_o \mid \theta) = p^\star(\mathbf{x}_o) \; \forall \mathbf{x}_o$ (Cannon et al., 2022). In contrast, we are not necessarily interested in reproducing the observed data $\mathbf{x}_o$ but only in inferring the parameter value $\theta$ from an observation $\mathbf{x}_o$. For this goal, naively using the standard definition is insufficient, as a model may be well-specified but still produce incorrect credible intervals for the parameters of interest $\theta$. This undesired behavior may happen, for example, if the model is over-parameterized, as illustrated in Appendix A.

Thus, in this work, we define model misspecification differently and align it with the setting motivated in Section 1. Intuitively, we describe model misspecification as the non-transferability of the posterior obtained from the simulator

to the prediction of real-world parameters. Formally, we assume that the pairs of parameters and observations $(\theta, \mathbf{x}_o)$ are i.i.d. from an unknown distribution $p^\star(\theta, \mathbf{x}_o)$, which implicitly defines $p^\star(\theta \mid \mathbf{x}_o)$, the Bayes optimal predictor of the parameter given an observation. Under this premise, we say a simulator is misspecified if $\exists \mathcal{S} \subseteq \Theta \times \mathcal{X} : \forall (\theta, \mathbf{x}_o) \in \mathcal{S}$,

$$p(\theta) = p^\star(\theta) \text{ and } p^\star(\theta \mid \mathbf{x}_o) \neq p(\theta \mid \mathbf{x}_s = \mathbf{x}_o).$$

Following this definition, we frame the problem of model misspecification in SBI as a learning task where our goal is to find a good estimator of $p^\star(\theta \mid \mathbf{x}_o)$. As we assume the simulator provides strong domain knowledge, we focus on the challenging settings where the dataset of labeled real observations $\mathcal{D} := \{(\theta^i, \mathbf{x}_o^i)\}_{i=1}^n$ that we have for learning $p^\star(\theta \mid \mathbf{x}_o)$ is small. In such settings, most examples must be saved for testing and only a small subset, denoted by the calibration set $\mathcal{C}$, remains available for training.

### 2.3 Semi-balanced Optimal Transport (OT)

As further motivated in Section 3, RoPE models the misspecification between simulations and real-world observations as an OT coupling. For readers unfamiliar with OT, a coupling between two distributions—e.g., $p(\mathbf{x}_s)$ and $p(\mathbf{x}_o)$—is a distribution $\pi^\star(\mathbf{x}_s, \mathbf{x}_o)$ on the product space whose marginals coincide with those two distributions while minimizing an expected cost $\mathbb{E}_{\pi^\star}[c(\mathbf{x}_o, \mathbf{x}_s)]$. The function $c : \mathcal{X}_o \times \mathcal{X}_s \to \mathbb{R}$ assigns a cost to any pair $(\mathbf{x}_o, \mathbf{x}_s) \in \mathcal{X}_o \times \mathcal{X}_s$.

In our setting, we can access a limited number $n_o$ of real-world observations $\{\mathbf{x}_o^i\}_{i=1}^{n_o}$, which we assume are i.i.d. from the unknown distribution $p^\star(\mathbf{x}_o)$. Writing $C := [c(\mathbf{x}_o^i, \mathbf{x}_s^j)]_{ij}$ for the cost matrix between observed and simulated data, we solve the discrete semi-balanced (Rabin et al., 2014) and entropy-regularized (Frogner et al., 2015) OT problem. This formulation preserves a strict marginal constraint on the observed data, but relaxes the marginal constraint on the simulated data, thus allowing certain simulations $\mathbf{x}_s$ to be discarded or down-weighted. Namely, given a set $\{\mathbf{x}_s^j\}_{j=1}^{n_s}$ of simulated observations, we search for the non-negative transport matrix $P^\star \in \mathcal{B}_o$ that satisfies the left marginal constraint,

$$\mathcal{B}_o = \left\{ P \in \mathbb{R}_+^{n_o \times n_s} : \sum_{j=1}^{n_s} P_{ij} = \frac{1}{n_o} \ \forall i = 1, ..., n_o \right\}$$

and solves

$$P^\star = \arg \min_{P \in \mathcal{B}_o} \langle P, C \rangle + \rho \, \mathrm{KL} \left( P^T \mathbf{1}_{n_o} \Big\| \frac{\mathbf{1}_{n_s}}{n_s} \right) + \gamma \langle P, \log P \rangle, \tag{2}$$

where $\mathbf{1}_n$ is a vector of ones with size $n$ and $\mathrm{KL}$ is the Kullback-Leibler divergence. Therefore, a larger $\rho > 0$ promotes a coupling that fits the marginal of simulated data more closely, and $\gamma > 0$ is a hyperparameter that encourages entropic transport matrices. This problem can be solved with a variant of the Sinkhorn algorithm (Cuturi, 2013) with efficient GPU implementations. In our experiments, we

rely on OTT (Cuturi et al., 2022) to return such a coupling $P^\star$, given the cost matrix $C$ and the parameters $\gamma$ and $\rho$, parameterized as $\tau = \rho/(\rho + \gamma)$. Setting $\tau = 1$ amounts to a perfectly balanced transport.

## 3 RoPE: Modeling Misspecification with OT

In this section, we formally introduce our robust posterior estimation algorithm (RoPE) and highlight some benefits of modeling misspecification with OT. RoPE approaches the problem of misspecification as a hybrid modeling task by combining the simulator with a misspecification model learned from the few observations in the calibration set. The main modeling assumption of RoPE is

$$\mathbf{x}_o \perp \theta \mid \mathbf{x}_s, \tag{3}$$

that is, given the simulated observations $\mathbf{x}_s$, the real observations $\mathbf{x}_o$ contain no additional information about the parameters $\theta$. As a consequence, we can express the posterior for real-world observations as $p(\theta \mid \mathbf{x}_o) = \int p(\theta \mid \mathbf{x}_s) p(\mathbf{x}_s \mid \mathbf{x}_o) d\mathbf{x}_s$, where $p(\theta \mid \mathbf{x}_s)$ is easily approximated with NPE. On the other hand, the conditional $p(\mathbf{x}_s \mid \mathbf{x}_o)$, which can be attributed to misspecification, is what RoPE intends to learn by estimating an OT coupling (that is then conditioned on $\mathbf{x}_0$, c.f. 4).

While this assumption introduces an information bottleneck, it does not prevent the method from achieving calibrated and informative posterior distributions—even if the assumption is only partially met in practice (e.g., tasks E and F in Figure 2). In fact, it acts as a regularizer, enabling the learning of a generalizable misspecification model from only a small calibration set, and it ensures that predictions remain grounded in the expert knowledge embedded in the simulator. This bottleneck can be limiting for simulators that are highly misspecified and fail to model the dependencies between parameters and observations. However, when the simulator encodes phenomena the practitioner believes to be invariant across different application environments, the assumption forestalls "shortcut learning" (Geirhos et al., 2020) from the calibration data and improves generalization. In Appendix D, we illustrate this property using real out-of-distribution data.

Intuitively, the discrete OT coupling $P^\star$ between the two point clouds $\{\mathbf{x}_s^i\}_{i=1}^{n_s}$ and $\{\mathbf{x}_s^i\}_{i=1}^{n_o}$ obtained from solving (2) can be seen as an approximation of a joint distribution $\pi^\star$ in $\mathcal{X}_o \times \mathcal{X}_s$ when $\tau = 1$ (see Appendix E for further discussion). Then, the modeled misspecification $\pi^\star$, together with our modeling assumption (3), defines the posterior distribution for real-world observations as

$$p(\theta \mid \mathbf{x}_o) = \int p(\theta \mid \mathbf{x}_s) \pi^\star(\mathbf{x}_s \mid \mathbf{x}_o) d\mathbf{x}_s, \tag{4}$$

where the posterior $p(\theta \mid \mathbf{x}_s)$ can be approximated very precisely with NPE (Papamakarios & Murray, 2016) as NFs are universal density estimators of continuous distributions (We-

henkel & Louppe, 2019; Draxler et al., 2024).

We approximate $\pi^\star$ by computing the OT coupling $P^\star$ between the test set $\mathcal{D}$ and a set $\{\mathbf{x}_s^j\}_{j=1}^{n_s}$ of $n_s$ simulations generated by running the simulator on parameters from the given prior $\theta^j \sim p(\theta)$. The cost function is defined in the next section. Thus, RoPE estimates the posterior for real-world observations as a mixture of the posteriors $\tilde{p}$ obtained with NPE, that is,

$$\tilde{p}(\theta \mid \mathbf{x}_o^i) := \sum_{j=1}^{n_s} \alpha_{ij}\tilde{p}(\theta \mid \mathbf{x}_s^j), \text{ where } \alpha_{ij} = n_o P_{ij}^\star. \quad (5)$$

### 3.1 Defining the OT Cost Function

In our setting, an ideal coupling would pair a real-world observation with simulations generated by the same parameters. Hence, the cost function should be insensitive to variation in the data (e.g., noise) that is independent of $\theta$. Formally, we can write $c(\mathbf{x}_o, \mathbf{x}_s) = c(\mathbf{h}_o(\mathbf{x}_o), \mathbf{h}_s(\mathbf{x}_s))$, where $\mathbf{h}_o$ and $\mathbf{h}_s$ are sufficient statistics for $\theta$ with respect to $\mathbf{x}_o$ and $\mathbf{x}_s$, respectively.

A key concern is to find a meaningful way to learn $\mathbf{h}_o$, the sufficient statistic for the real observations. As discussed in Appendix G, we can learn an approximate minimal sufficient statistic $\mathbf{h}_{\omega^\star}$ for the simulated observations with NPE. Because the simulator carries information about the true generative process, our approach is to fine-tune $\mathbf{h}_{\omega^\star}$ using the calibration set, which is otherwise too small to learn a representation from real-world data only. Denoting the fine-tuned neural network as $\mathbf{g}_\varphi : \mathcal{X}_o \to \mathbb{R}^l$, the fine-tuning objective reads

$$\mathcal{L}(\varphi;\mathcal{C}) := \sum_{i=1}^{n_c} |\mathbf{g}_\varphi(\mathbf{x}_o^i) - \mathbb{E}_{\varepsilon\sim\mathcal{U}[0,1]}[\mathbf{h}_{\omega^\star}\left(S(\theta^i,\varepsilon)\right)]|_2, \quad (6)$$

where the expectation is approximated via a Monte-Carlo approximation. The training of $\mathbf{g}$ starts from the weights $\omega^\star$ and optimizes (6) with gradient descent. Optimizing (6) enforces, at least on the calibration set, that $\mathbf{g}$ and $\mathbf{h}$ are close in L2 norm when applied to observations from the same parameter $\theta$. Thus, we define the OT cost as $c(\mathbf{x}_o, \mathbf{x}_s) := |\mathbf{g}_{\varphi^\star}(\mathbf{x}_o) - \mathbf{h}_{\omega^\star}(\mathbf{x}_s)|_2$, where $\mathbf{g}_{\varphi^\star}$ is the NSE obtained after fine-tuning (6). Figure 4 in Appendix B depicts RoPE's training and inference steps. We discuss the computational cost of RoPE in Section H.

### 3.2 On the benefits of using optimal transport to handle misspecification

While we could have chosen other approaches to model $p(\mathbf{x}_s \mid \mathbf{x}_o)$—e.g., conditional deep generative models—several attractive properties directly follow from modeling the misspecification as an OT coupling between simulated and real-world measurements. First, **a self-calibration property**: by modeling the posterior as (5), when $\tau = 1$ (i.e., the transport is perfectly balanced), the marginal pos-

terior distribution over the test set, i.e., $\tilde{p}(\theta) := \int \tilde{p}(\theta \mid \mathbf{x}_o)p^\star(\mathbf{x}_o)\mathrm{d}\mathbf{x}_o$, converges to the prior distribution as the number of simulated observations $N_s$ approaches infinity, as expected from a well-estimated posterior distribution. A proof and further discussion of this self-calibration property is given in Section F. Second, **a control mechanism for the posteriors' confidence**: the entropic regularization of OT not only enables fast computation of the transport coupling but also provides an effective control mechanism to balance the calibration of the posterior with its informativeness. Indeed, for small entropic regularization, the estimated posteriors have low entropy and may be overconfident, as they are sparse mixtures of a few simulation posteriors $\tilde{p}(\theta \mid \mathbf{x}_s^j)$. In contrast, for large values of $\gamma$ in (2), the coupling matrix becomes uniform and the corresponding posteriors tend to the prior, as $p(\theta \mid \mathbf{x}_o) \approx \frac{1}{n_s}\sum_j^{n_s} \tilde{p}(\theta \mid \mathbf{x}_s^j)$ is a Monte-Carlo approximation of $\mathbb{E}_{p(\mathbf{x}_s)}[\tilde{p}(\theta \mid \mathbf{x}_s)] \approx p(\theta)$. Thus, the practitioner can optimize the hyper-parameter $\gamma$ to find the right trade-off between calibration of the estimated posteriors, favored by higher $\gamma$, and their informativeness, favored by lower $\gamma$. Finally, **robustness to prior misspecification**: by enabling the transport to be unbalanced—that is, to discard simulated observations when $\tau < 1$—RoPE can flexibly depart from the assumed marginal distribution of $p(\theta)$ and be robust to prior misspecification. Thus, the parameter $\tau$ can be seen as a control mechanism to account for the user's confidence in the prior distribution. In the rest of the text, we denote the method as RoPE$^\star$ when $\tau < 1$ and as RoPE when $\tau = 1$. In Section 5.1, we provide guidance on how to set $\gamma$ and $\tau$ in practice.

## 4 Related Work

The problem we address shares fundamental similarities with sim2real transfer learning, where the goal is to bridge the gap between simulated and real-world data. In robotics and computer vision, this challenge has been tackled through domain randomization (Tobin et al., 2017), which increases simulation diversity to improve real-world generalization, and domain adaptation techniques (Ganin et al., 2016; Long et al., 2015; Bousmalis et al., 2018) that learn domain-invariant representations. However, unlike these approaches that typically focus on point predictions, RoPE addresses the more challenging problem of transferring uncertainty quantification from simulation to reality while preserving calibration properties.

The setting we consider also naturally connects to semi-supervised learning (Zhu, 2005), as both involve leveraging abundant unlabeled data alongside limited labeled examples. Our setup with the calibration set resembles few-shot learning scenarios (Wang et al., 2020), where rapid adaptation occurs with minimal labeled examples. While classical semi-supervised methods focus on exploiting unlabeled data for classification or regression tasks, our approach differs in

that it leverages a large set of labeled data obtained through simulation. Crucially, unlike standard semi-supervised or few-shot learning, where labeled and unlabeled data come from the same distribution, we must explicitly account for the distributional mismatch between simulated and real observations.

In both likelihood-based and simulation-based inference settings, model misspecification has recently gained substantial interest from the research community. Among developed strategies, works that take inspiration from generalized Bayesian inference (Bissiri et al., 2016) are numerous (Dellaporta et al., 2022; Chérief-Abdellatif & Alquier, 2020; Matsubara et al., 2022; Pacchiardi & Dutta, 2021; Schmon et al., 2020; Gao et al., 2023; Frazier et al., 2023). In the specific context of SBI, recent works (Ward et al., 2022; Huang et al., 2023; Kelly et al., 2023) have investigated solutions to improve the robustness of existing neural-network-based SBI methods to model misspecification, detecting it at inference time (Schmitt et al., 2023). Similarly, Frazier et al. (2020) studied the impact of misspecification on approximate Bayesian computation methods (ABC, Rubin, 1984), introducing diagnostics to detect it and proposing strategies to make ABC robust. For the interested reader, Nott et al. (2023) review restricted likelihood methods, Bayesian modular inference, and parametric projection methods, which are standard frameworks to handle model misspecification in likelihood-based Bayesian inference.

In contrast to these approaches, we frame model misspecification in SBI as a learning problem, recognizing that if the ultimate goal is to perform inference over parameters for downstream decision-making, it is essential to have a test set to empirically validate the performance of any inference procedure. RoPE leverages a small subset of this test set as a calibration set to overcome the modeled misspecification in a supervised manner.

## 5 Experiments

Our experiments aim to (1) empirically validate the discussion in Section 3.2, and (2) illustrate settings in which RoPE enables uncertainty quantification under model misspecification and small calibration datasets. The experiments comprise two existing benchmarks from the SBI literature, two synthetic benchmarks, and two new benchmarks from real physical systems for which both labeled data and simulators are available. While these benchmarks remain simplified versions of real-world scenarios, they represent various types of misspecification with varied parameter and observation spaces, allowing us to study RoPE's performance under diverse configurations. We briefly describe each task and provide examples of real vs. simulated observations in Figures 1 and 2. Further details about the experimental setup can be found in Appendix I.

**Task A & B (synthetic): CS & SIR .** We reproduce the cancer and stromal cell development (CS) and the stochastic epidemic model (SIR) benchmarks from Ward et al. (2022). We provide a description of the parameters, observations and synthetic misspecification in Appendix I.1

**Task C (synthetic): Pendulum.** The damped pendulum is a common benchmark for hybrid learning algorithms (Takeishi & Kalousis, 2021; Yin et al., 2021; Wehenkel et al., 2022) that leverage both domain knowledge and real-world data. The simulator outputs the horizontal position of a frictionless pendulum given its fundamental frequency $\omega_0 \in \mathbb{R}^+$ and amplitude $A \in \mathbb{R}^+$, with randomness introduced via a phase shift and white measurement noise. As misspecified "real-world" data, we generate observations from a damped pendulum with friction.

**Task D (synthetic): Hemodynamics.** Following Wehenkel et al. (2023), we define the task of inferring the stroke volume (SV) and the left ventricular ejection time (LVET) from normalized arterial pressure waveforms. The simulator is a PDE solver (Melis, 2017) that produces an 8-second time-series $\mathbf{x}_s$ sampled at 125Hz. As synthetic misspecification, the simulator assumes all arteries have constant length, whereas this parameter varies in the "real-world" data.

**Task E (real): Light Tunnel.** We employ one of the light tunnel datasets from Gamella et al. (2025). The tunnel is an elongated chamber with a controllable light source at one end, two linear polarizers mounted on rotating frames, and a camera. Our task consists of predicting the color setting of the light source $((R, G, B) \in [0, 255]^3)$ and the dimming effect of the polarizers $\alpha \in [0, 1]$ from the captured images. The simulator takes the parameters $\theta := [R, G, B, \alpha]$ and produces an image consisting of a hexagon roughly the size of the light source, with a color equal to $[\alpha R, \alpha G, \alpha B]$.

**Task F (real): Wind Tunnel.** We employ one of the wind tunnel datasets from Gamella et al. (2025). The tunnel is a chamber with two controllable fans that push air through it, and barometers that measure air pressure at different locations. A hatch controls the area of an additional opening to the outside. The dataset is a collection of pressure curves that result from applying a short impulse to the intake fan power and measuring the change in air pressure inside the tunnel. Our inference task consists of predicting the hatch position, $\theta := H \in [0, 45]$ given a pressure curve. As a simulator model, we adapt the physical model given in Gamella et al. (2025, Appendix IV).

**Metrics.** We consider two metrics to assess whether RoPE provides reliable and useful uncertainty quantification. First, given a labeled test set $\{(\theta^i, \mathbf{x}_o^i)\}_{i=1}^n$, we compute the log-posterior probability (LPP) as LPP $:= \frac{1}{n} \sum_{i=1}^n \log \tilde{p}(\theta^i \mid \mathbf{x}_o^i) \approx \mathbb{E}_{p(\theta, \mathbf{x}_o)} [\log \tilde{p}(\theta \mid \mathbf{x}_o)]$. The LPP, also called the negative log probability of the true test parameter (NLTP), is

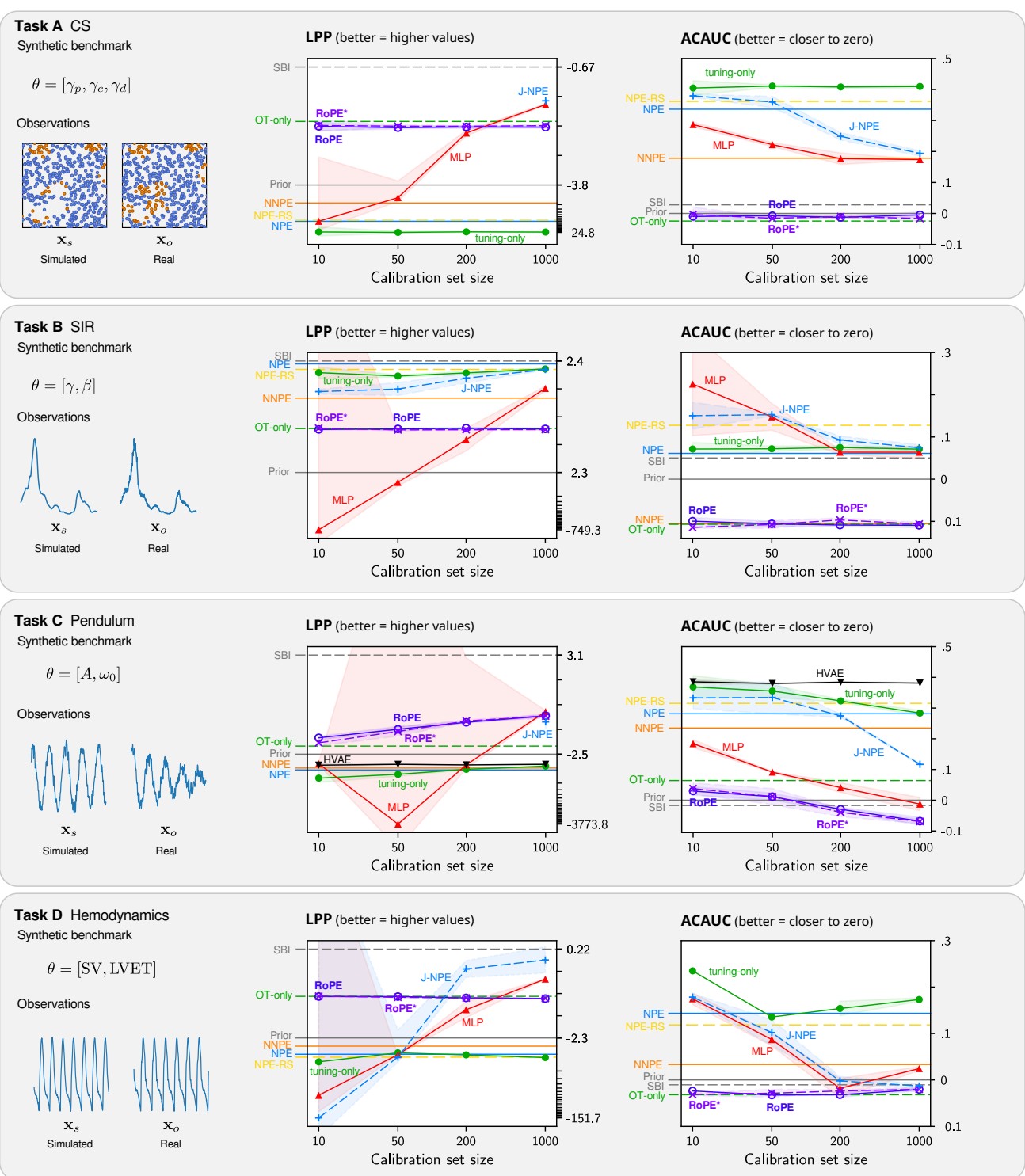

Figure 1: Results for our method (RoPE) and the competing baselines on six benchmark tasks. For each task, we show an example of the real observations ($\mathbf{x}_o$) and the observations produced by the misspecified simulator ($\mathbf{x}_s$). We show each method's LPP and ACAUC metrics, as computed on a labeled test set of size 2000. Horizontal lines without markers correspond to the methods that do not use the calibration set, producing a constant score. We report the average metrics and $\pm 1$ std. deviation over three random draws of the test set and additional sources of randomness. In some instances, e.g., J-NPE or NPE-RS in task C, the likelihood can be $-\infty$ and is not plotted. For readability of the LPP metric, we use a linear scale between the SBI and the Prior and a logarithmic scale for values below that.

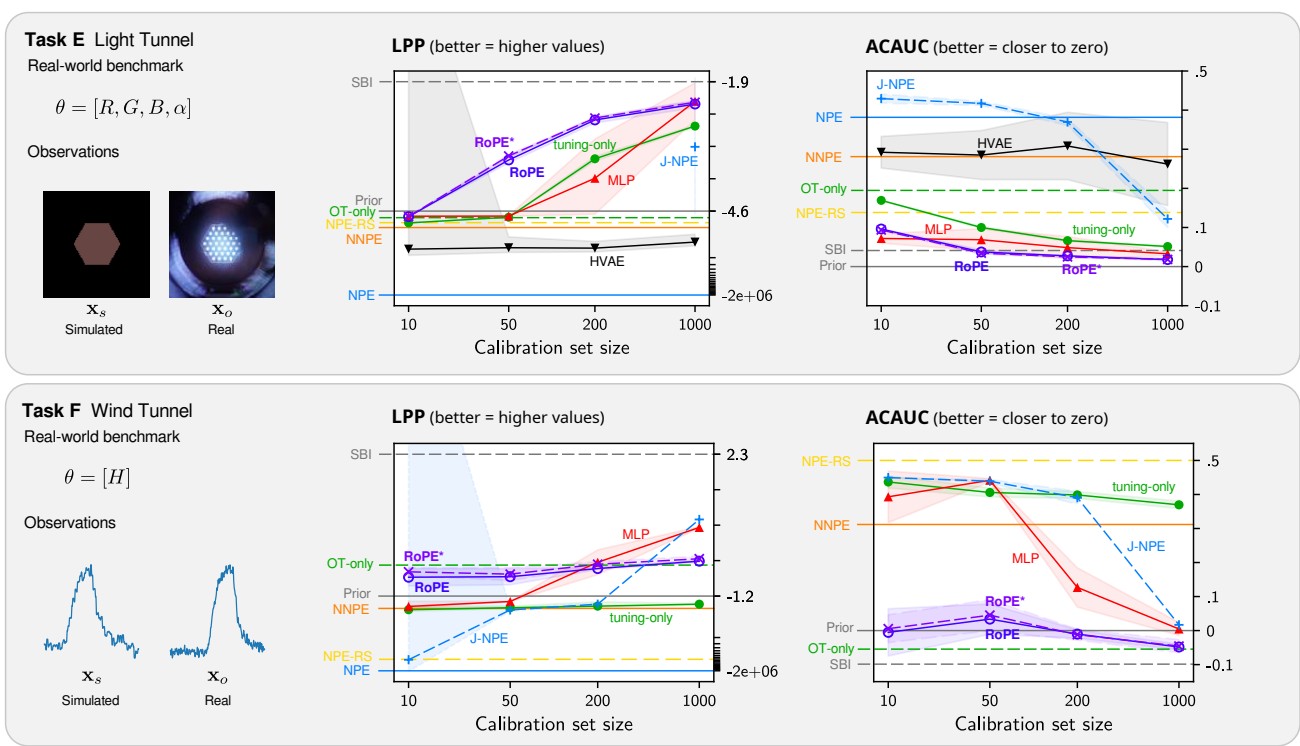

Figure 2: Continuation of Figure 1 above. For task F, the ACAUC of the NPE baseline is -0.5 and not shown.

an empirical estimation of the expectation over possible observations of the negative cross entropy between the true and estimated posterior; thus, for an infinite test set, it is only maximized by the true posterior. LPP characterizes the entropy reduction on the estimation of $\theta$ achieved by a posterior estimator $\tilde{p}$ when given one observation, on average, over the test set. Second, the average coverage AUC (ACAUC) indicates the average calibration of $k$ 1D credible intervals extracted from the estimated posteriors, i.e., $\text{ACAUC} := \frac{1}{kn} \sum_{j=1}^{k} \sum_{i=1}^{n} \int_{0}^{1} \alpha - \mathbf{1}[\theta_j^i \in \Theta_{\tilde{p}(\theta_j|\mathbf{x}_o^i)}(\alpha)]d\alpha$, where $\Theta_{\tilde{p}(\theta_j|\mathbf{x}_o^i)}(\alpha)$ denotes the credible interval for the j-th dimension of the parameter $\theta$ at level $\alpha$. Its value is positive (resp. negative) if, on average over different credible levels, parameter dimensionality, and observations, the corresponding credible intervals are overconfident (resp. underconfident). The ACAUC of a perfectly specified prior distribution is zero. The integral can be efficiently approximated, as described in Appendix J. ACAUC does not capture joint calibration, as dependencies between parameters are not explicitly assessed. Alternative dependence-sensitive metrics may require larger test sets to be stable. For all experiments, we compute the LPP and ACAUC on labeled test set containing 2000 pairs $(\theta, \mathbf{x}_o)$.

**Baselines.** As a sanity check, we compare the performance of RoPE against four reference baselines: the **prior** $p(\theta)$, which amounts to the lower bound on the LPP for any calibrated posterior estimator when the prior is well-specified; the **SBI** posterior, which is an NPE trained and tested on simulated data and thus provides an upper bound on the LPP for RoPE under the independence assumption $\mathbf{x}_o \perp \theta \mid \mathbf{x}_s$ (see Appendix I for more details); (**NPE**) a posterior estimator fitted to the simulated data and applied to the real data; and (**J-NPE**) a posterior estimator trained jointly on the pooled simulated and real observations. The latter two baselines represent some first approaches that a practitioner may consider. Furthermore, to asses how a fully supervised approach would fare if trained directly on the calibration set, we compare the performance of RoPE to **MLP**, which trains a neural network to predict the mean and log-variance of a Gaussian posterior distribution by maximizing the calibration set log-likelihood. We train both the MLP and J-NPE baselines in a supervised way, and we thus expect these baselines to perform strongly as the size of the calibration set becomes sufficiently large and the test data is i.i.d. We also run **NPE-RS** (Huang et al., 2023), which trains a robust version of NPE with a regularization loss that forces the distributions of NSE on simulated and test data to match. For a fair comparison with RoPE, we use the $n = 2000$ test examples to compute the regularization, informing NPE-RS as much as possible. We additionally run Noisy NPE (**NNPE**, Ward et al., 2022), the amortized version of RNPE introduced in the same paper, which improves the robustness of NPE by introducing a Spike and Slab error model on simulated data statistics. We also run **HVAE** (Takeishi & Kalousis, 2021), which constitutes a strong baseline when the simulator can be made differentiable (tasks C and E) but is not directly applicable otherwise. More details about the

experimental setup can be found in Appendix I.

## 5.1 Results

Figure 1 compares the performance of RoPE and the other methods and baselines on the six tasks we consider with a correctly specified prior. To demonstrate that applying RoPE is straightforward, we deliberately fix $\gamma = 0.5$ for RoPE and $\tau = 0.9$ for RoPE$^\star$ in all tasks. In Figure 3, we further study the role of these hyperparameters in optimizing performance.

**RoPE achieves robust posterior estimation for all tasks.** As mentioned above, the SBI and prior baselines provide upper and lower bounds on the expected performance of a well-calibrated posterior estimator, under the modeling assumption made in Section 3. For all tasks, even with minimal calibration budgets, RoPE is the only method that consistently returns well-calibrated, or sometimes slightly under-confident, posterior estimates while significantly reducing uncertainty compared to the prior distribution. As the size of the calibration set increases, we see that J-NPE and MLP adapt and their performance improves and aligns with or outperforms RoPE. This adaptability is an expected behavior in i.i.d. settings, where real-world data eventually allows finding the minimizer of empirical risk among a class of predictors. Nevertheless, these two baselines tend to be overconfident even for larger calibration sets, as highlighted by their positive ACAUC numbers, which are significantly larger than RoPE's in almost all configurations. Moreover, on task E, where posteriors are complex conditional distributions—whose entropy increases with darker images and contain non-trivial dependencies between parameters—RoPE remains the best approach, even with a calibration set containing more than $1000$ examples. As an outlier, we observe that NPE trained on simulated data achieves the best results for the SIR benchmark (Task B), indicating that the misspecification of this benchmark is not a challenging test case for existing SBI methods and may not be a meaningful test for methods that cope with model misspecification. Finally, because interpreting these metrics can be difficult, we complement these numerical results with corner and calibration plots for all tasks in Appendix K.

**Ablation study.** RoPE combines two steps with distinct roles, shown in Figure 4, Appendix B: (1) a fine-tuning step, which improves the domain generalization of the NSE; and (2) an OT step, aiming to model the misspecification as a coupling between simulations and observations. To better understand their respective contribution to the performance of RoPE, we look at two ablated versions of our algorithm: **tuning-only** which appends the fine-tuned NSE to the NF trained on simulated data and directly applies it to the real observations without an OT step; and **OT-only**, which directly performs OT with L2-norm in the original NSE space $c(\mathbf{x}_o, \mathbf{x}_s) = |\mathbf{h}_{\omega^\star}(\mathbf{x}_o) - \mathbf{h}_{\omega^\star}(\mathbf{x}_s)|_2$. In Figure 1,

we observe that the results for tuning-only are poor except for Task B, where misspecification is negligible. In contrast, for tasks A, D, and F, OT-only exhibits performance on par with RoPE. Nevertheless, RoPE can significantly outperform OT-only, such as in tasks C and E where the misspecification is significant. We conclude that the OT step is crucial and fine-tuning is sometimes necessary. In practice, we recommend to first evaluate the performance of OT-only on the test set, and optimize $\gamma$ before using a subset of the test samples for fine-tuning.

**Effect of entropic regularization—setting $\gamma$.** In Figure 3a, we study the effect of entropic regularization by varying the regularization parameter $\gamma$. For all values of $\gamma$, excluding $\gamma \geq 5$, we observe that both LPP and ACAUC consistently improve with the calibration set size. For large values of $\gamma$, the entropic regularization dominates and pushes toward a uniform mapping, resulting in posteriors that approximate the prior distribution and are barely affected by the calibration set size. These empirical results are consistent with the theoretical discussion in Subsection 3.2. As a recommendation for practitioners, our empirical evaluation suggests that values between $0.1$ and $1$ provide well-calibrated and precise credible intervals. Ideally, the practitioner shall keep a portion of the calibration set for validation, using it to optimize $\gamma$ based on the metrics of interest. If this is not possible, we recommend employing $\gamma = 0.5$, which offers sharp and calibrated posteriors on all our benchmarks.

**RoPE$^\star$ for prior misspecification—setting $\tau$.** We now study the impact of prior misspecification on RoPE and its unbalanced version RoPE$^\star$. In Figure 3, we compare the performance of RoPE ($\gamma = 0.5$ and $\tau = 1$) and RoPE$^\star$ ($\gamma = 0.5$ and varying $\tau$) on extensions of Task E and C, where the ground-truth parameters of the test dataset come from distributions different to the assumed prior distributions. For task E, we observe that RoPE's performance is robust to the prior misspecification; it provides well-calibrated and informative posteriors, as is also visible in the corner plots of Figure 5 in Appendix C. While the gap between RoPE and RoPE$^\star$ is negligible in the case of a well-specified prior (see Task E in Figure 1), under prior misspecification RoPE$^\star$ leverages the additional flexibility in the OT solution and discards some of the simulated observations, achieving higher LPP. Similarly, for Task C in Figure 3c, when there is no prior misspecification, RoPE (i.e, $\tau = 1$) achieves the best performance; using lower values of $\tau$ becomes preferable as prior misspecification increases. From these experiments, we recommend leveraging $\tau$ as a hyperparameter describing the confidence in the assumed prior distribution—setting its value to $0.9$ offers robust performance for both well-specified and partially misspecified priors. The user shall also explore lower values when there is suspicion that the prior distribution is overly spread with respect to the correct prior.

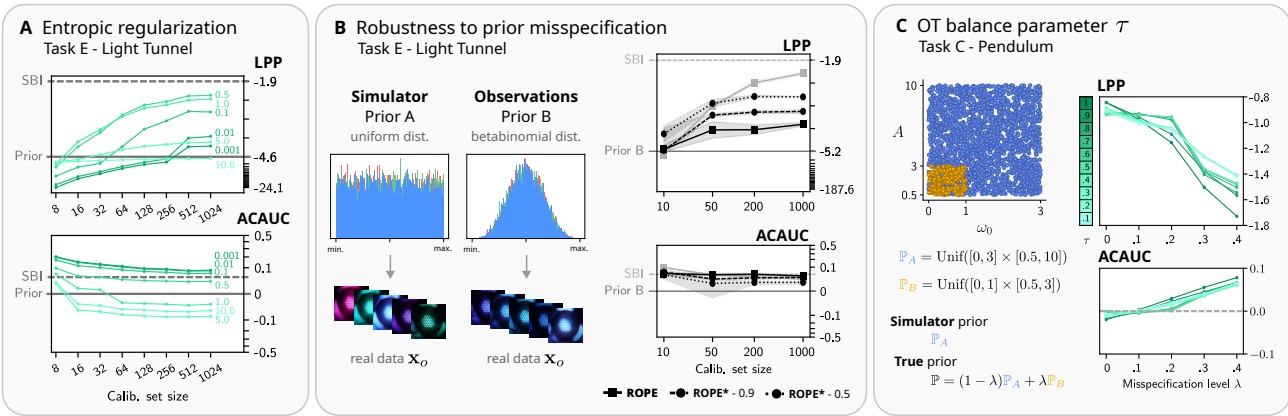

Figure 3: (a) Effect of $\gamma$ on the LPP and ACAUC scores of RoPE on the light-tunnel task for different sizes of the calibration set. The value of $\gamma$ is shown by each curve. For reference, we plot the metrics achieved by the SBI posterior and prior distribution on simulated data. (b-c) Effect of $\tau \in [0.1, 1]$ under a prior misspecification in Task E (b); and for various levels of prior misspecification in task C (c).

## 6   Discussion

While Section 5 demonstrates the effectiveness of RoPE, opportunities for future work remain, which we discuss now.

**Curse of dimensionality.** While our experiments focused on low-dimensional parameter spaces, as is common for many applications of SBI, the dimensionality of $\theta$ may impact two critical parts of RoPE. First, with each additional parameter $\theta_{k+1}$, given $\mathbf{x}_o$, the NSE must encode up to $K$ dependencies between $\theta_{k+1}$ and the other dimensions $\theta_1, \ldots, \theta_k$. While generating more simulations can address the curse of dimensionality in the simulation space, fine-tuning on a small calibration may no longer suffice to cope with misspecification. Second, the dimensionality of the manifold on which the NSE projects the simulated and real-world observations will grow, and finding a meaningful coupling between the two populations may require larger sample sizes. A potential solution is to focus on marginal or 2D posterior distributions and ignore higher-dimensional dependencies in $p(\theta \mid \mathbf{x}_o)$. Nevertheless, extending RoPE to such settings certainly opens new questions, e.g., concerning the development of better fine-tuning strategies that can leverage calibration sets with incomplete labels.

**Non-iid Calibration Sets.** An important assumption made by RoPE is that the calibration set contains i.i.d. samples drawn from the same distribution $p^\star(\theta, \mathbf{x}_o)$ as the test data. However, practical constraints may lead to calibration data being collected from a different, potentially biased, distribution $\tilde{p}(\theta, \mathbf{x}_o)$. We identify two main scenarios. If $\tilde{p}$ and $p^\star$ share the same support, the fine-tuning step can still correct for the distributional shift, especially with a sufficiently large calibration set. For smaller sets, RoPE's robustness hinges on the neural statistic estimator's (NSE) ability to generalize. Moreover, the optimal transport (OT) step provides additional resilience: observations where the fine-tuned NSE performs well will be accurately matched, leading to reliable posteriors, while poorly generalized ob-

servations may cause the posterior to revert to the prior. In the more challenging scenario where $\tilde{p}$ and $p^\star$ have disjoint support, even arbitrarily large calibration sets may fail to provide relevant training examples, making fine-tuning highly dependent on out-of-distribution generalization. Here, the OT step is expected to highlight this issue, as the lack of meaningful matches will cause the transport matrix to become uniform, leading the posterior to revert to the prior. Appendix L further investigate RoPE's sensitivity to these practical challenges, on the Light Tunnel task, using a calibration set from a different prior than the test set, approximating the 'same support' scenario.

**Other extensions.** Similar to incomplete labels, in certain applications we may only have access to noisy labels, measured with a well-modeled but noisy measurement process. Further developing the fine-tuning stage to exploit such noisy labels would be necessary to make an approach similar to RoPE applicable. Our strategy of modeling misspecification as an OT coupling opens up several avenues to address more specific problem setups. For example, we can leverage the inductive bias in the neural network architecture of neural OT to better cope with large test sets. This appears as a promising direction to amortize the mapping between simulation and real-world data.

**Conclusion.** Motivated by important applications where SBI is not applied due to its sensitivity to model misspecification, we have introduced RoPE, a method that jointly exploits a calibration set and optimal transport to extend neural posterior estimation for misspecified simulators. Our experiments on diverse benchmarks demonstrate RoPE's ability to estimate calibrated and informative posterior distributions for various simulators and real-world examples. Overall, we have framed model misspecification as a challenge in transferring predictive models from simulated to real-world data. Our work highlights the need for a labeled test set to validate inference quality, encouraging future research to treat misspecification as a machine learning problem.

## Acknowledgements

The authors would like to acknowledge Michal Klein for his help with OTT library and Maria Cervera, Laura Manduchi, Joe Futoma, Andy Miller and Pierre Ablin for providing useful feedback on the manuscript.

## Impact statement

This paper presents a framework and an algorithm to address model misspecification in simulation-based inference (SBI). SBI is predominantly applied in scientific fields where complex simulators of physical phenomena are available, such as astronomy, medicine, particle physics, or climate modeling. A priori, this circumscribes the application of our algorithm to highly specialized scientific domains in the natural sciences, precluding issues such as fairness or privacy. However, its application to the scientific domain is not exempt from societal or ethical implications, particularly when computer simulations may inform research or policy decisions. In this regard, we find some properties of the algorithm particularly promising, such as uncertainty quantification and the limitation of not drawing conclusions beyond the given expert model. However, more work is needed to deeply understand the reliability of these properties and how they are affected by violations of the core assumptions, such as a well-specified prior. Such work should precede any sort of over-selling to practitioners about the benefits of the algorithm. Rather, we see our work as a contribution towards a more broad and successful application of SBI techniques; success in this endeavor, as for the establishment of any scientific tool, will require an iterative dialogue between the scientists who develop the methodology and those who use it.

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

# A Model misspecification

## A.1 Mis-calibration vs Misspecification

To further elucidate the distinction between posterior calibration and model misspecification, it is essential to highlight their respective scopes and the specific challenges they address.

Posterior calibration focuses on ensuring that the predicted posterior distributions accurately reflect the true uncertainty in parameter estimates given the observations, under the assumption that the simulator is well-specified. Methods such as those proposed by Falkiewicz et al. (2024); Delaunoy et al. (2022) address this by improving the alignment between the expected and actual coverage probabilities of the posterior. These approaches generally assume that the simulator faithfully represents the generative process of the observed data, enabling calibration to be evaluated and improved by leveraging simulations. While important, these methods do not account for discrepancies between the simulator and real-world data, which are precisely the scenarios we target in this work.

Model misspecification, on the other hand, arises when the simulator fails to capture the true generative process underlying the observed data. This results in systematic discrepancies that cannot be corrected solely by optimizing posterior calibration techniques. Misspecification introduces a gap between the simulated and real-world distributions, and this gap is only observable when real-world data is available. Unlike posterior calibration, addressing misspecification requires methods that can robustly leverage the simulator despite its inaccuracies, while incorporating real-world observations to mitigate the impact of the mismatch.

In our work, we explicitly focus on handling model misspecification. This distinction is reflected in the design of our approach and the evaluation scenarios we consider, such as Task E, where the simulated data diverges significantly from the real-world measurements. While posterior calibration methods may perform well in a well-specified context, they are not designed to cope with such gaps. Instead, we prioritize creating predictive models that balance informativeness and robustness in the presence of misspecification, even if achieving perfect calibration remains an open and challenging problem.

## A.2 Comparison between model misspecification definitions

We provide a toy example to show how a simulator may be well-specified according to the standard definition of misspecification but still provide biased estimates of the target parameter when applied to real data.

Consider the following setting: a noisy sensor measures some physical quantity $\theta$, producing measurements $\mathbf{x}_o^1, \ldots, \mathbf{x}_o^n \overset{\text{i.i.d.}}{\sim} \mathbb{P}^\star$, where $\mathbb{P}^\star := \mathcal{N}(\theta^\star, 1)$ is a normal distribution centered around the 'true' value $\theta^\star$. Let $\{\mathbb{P}_\theta : \theta \in \mathbb{R}\}$ be a simulator of this process with $\mathbb{P}_\theta := \mathcal{N}(\mu, 1)$, where $\mu := \theta + \lambda$ and $\lambda > 0$ is a fixed scalar constant, which is a misspecification in the simulator that falsely accounts for a non-existing offset in the sensor that produced the real observations $\mathbf{x}_o^1, \ldots, \mathbf{x}_o^n$.

According to the standard definition of misspecification, the simulator is well specified, as setting $\theta \leftarrow \theta^\star - \lambda$ yields $\mathbb{P}_\theta = \mathbb{P}^\star$. However, the posterior estimates we obtain with this simulator are biased with respect to the true parameter $\theta^\star$.

To see this, let us compute the posterior under a Gaussian prior $\mathcal{N}(\theta^\star, 1)$ over the parameter $\theta$, centered on the true value $\theta^\star$.

Taking advantage of the conjugate prior, the posterior $p(\theta \mid \mathbf{x}_o^1, \ldots, \mathbf{x}_o^n)$ becomes

$$
\begin{aligned}
p(\theta \mid \mathbf{x}_o^1, \ldots, \mathbf{x}_o^n) &\propto p(\theta)p(\mathbf{x}_o^1, \ldots, \mathbf{x}_o^n \mid \theta) \\
&= p(\theta) \prod_{i=1}^{n} p(\mathbf{x}_o^i \mid \theta) \\
&= \frac{1}{\sqrt{2\pi}} \exp\left(-\frac{1}{2}(\theta - \theta^\star)^2\right) \prod_{i=1}^{n} \frac{1}{\sqrt{2\pi}} \exp\left(-\frac{1}{2}(\mathbf{x}_o^i - \mu)^2\right) \\
&\propto \exp\left(-\frac{1}{2}(\theta - \theta^\star)^2 - \frac{1}{2}\sum_{i=1}^{n}(\mathbf{x}_o^i - \mu)^2\right) \\
&= \exp\left(-\frac{1}{2}\left[\theta^2 + (\theta^\star)^2 - 2\theta\theta^\star + \sum_{i=1}^{n}(\mathbf{x}_o^i)^2 + n\mu^2 - 2\mu\sum_{i=1}^{n}\mathbf{x}_o^i\right]\right) \\
\text{(drop const. terms)} \quad &\propto \exp\left(-\frac{1}{2}\left[\theta^2 - 2\theta\theta^\star + n\mu^2 - 2\mu\sum_{i=1}^{n}\mathbf{x}_o^i\right]\right) \\
(\mu = \theta + \lambda) \quad &= \exp\left(-\frac{1}{2}\left[\theta^2 - 2\theta\theta^\star + n\theta^2 + n\lambda^2 + 2n\lambda\theta - 2\theta\sum_{i=1}^{n}\mathbf{x}_o^i - 2\lambda\sum_{i=1}^{n}\mathbf{x}_o^i\right]\right) \\
\text{(drop const. terms)} \quad &\propto \exp\left(-\frac{1}{2}\left[\theta^2 - 2\theta\theta^\star + n\theta^2 + 2n\lambda\theta - 2\theta\sum_{i=1}^{n}\mathbf{x}_o^i\right]\right) \\
&= \exp\left(-\frac{1}{2}\left[(n+1)\theta^2 - 2\theta(\theta^\star - n\lambda + \sum_{i=1}^{n}\mathbf{x}_o^i)\right]\right) \\
&= \exp\left(-\frac{1}{2(n+1)^{-1}}\left[\theta^2 - 2\theta\left(\frac{1}{n+1}\right)(\theta^\star - n\lambda + \sum_{i=1}^{n}\mathbf{x}_o^i)\right]\right) \\
\text{(complete square)} \quad &\propto \exp\left(-\frac{1}{2(n+1)^{-1}}\left[\theta - \left(\frac{1}{n+1}\right)(\theta^\star - n\lambda + \sum_{i=1}^{n}\mathbf{x}_o^i)\right]^2\right),
\end{aligned}
$$

that is, a normal distribution $\mathcal{N}(\tau, \gamma^2)$ with mean

$$
\tau = \left(\frac{1}{1+n}\right)\left(\theta^\star - n\lambda + \sum_{i=1}^{n}\mathbf{x}_o^i\right)
$$

and variance $\gamma^2 = (n+1)^{-1}$. Thus, the posterior is biased, e.g., the posterior mean $\tau$ is a biased estimator of $\theta^\star$ with $\mathbb{E}[\theta^\star - \tau] = \theta^\star - \lambda\left(\frac{n}{n+1}\right)$.

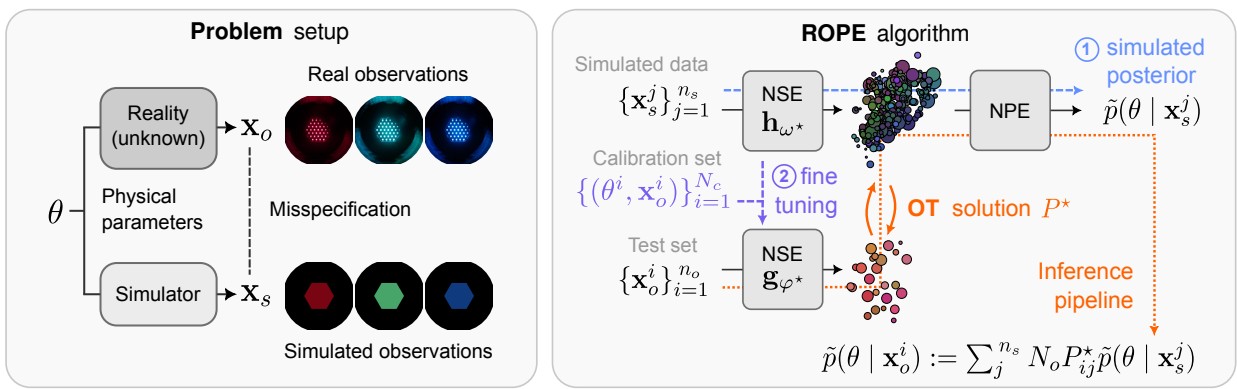

Figure 4: (*left*) Problem setup: we consider a real-world process which depends on some physical parameters $\theta$. Given real observations $\mathbf{x}_o$ of the process, our goal is to provide uncertainty quantification on the underlying parameters $\theta$. To help us, we have access to a misspecified simulator that takes parameters $\theta$ as input and produces simulated observations $\mathbf{x}_s$. (*right*) A visualization of RoPE. The training consists of two steps: (1) given the simulated data, we approximate the posterior using NPE, resulting in the NSE $\mathbf{h}_{\omega^\star}$; (2) using the calibration set, we fine-tune $\mathbf{h}_{\omega^\star}$ into $\mathbf{g}_{\varphi^\star}$ using the objective (6). At test time, we solve the optimal transport (OT) problem between the representations $\{\mathbf{h}_{\omega^\star}(\mathbf{x}_s^j)\}_{j=1}^{n_s}$ and $\{\mathbf{g}_{\varphi^\star}(\mathbf{x}_o^i)\}_{i=1}^{n_o}$, resulting in our estimated posterior (5), the average of simulations' posteriors weighted by the OT solution $P^\star$. See Algorithm 1 in Appendix B for more details.

# B   The RoPE Algorithm

---

**Algorithm 1** Posterior Inference using Robust Neural Posterior Estimation (RoPE)

---

**Input:** Simulator $S(\theta, \varepsilon)$, prior distribution $p(\theta)$, calibration set $\mathcal{C} = \{(\mathbf{x}_o^i, \theta^i)\}_{i=1}^{N_c}$, test set $\mathcal{D} = \{\mathbf{x}_o^i\}_{i=1}^{N_o}$

**Output:** $\tilde{p}(\theta \mid \mathbf{x}_o) \forall \mathbf{x}_o^i \in \mathcal{D}$

**Step 1: Neural Posterior Estimation (NPE)**

Train neural network $\mathbf{h}_\omega$ and conditional normalizing flow $p(\theta \mid \cdot)$ using NPE:

$$\tilde{p}, \omega^\star = \arg\max_{p,\omega} \mathbb{E}_{\substack{\theta \sim \pi(\theta) \\ \varepsilon \sim \mathcal{U}[0,1]}} \left[ \log p(\theta \mid \mathbf{h}_\omega(S(\theta, \epsilon))) \right]$$

**Step 2: Fine-tune sufficient statistics $\mathbf{h}_{\omega^\star}$ on the Calibration Set**

$\mathbf{g}_\psi := \mathrm{COPY}(\mathbf{h}_{\omega^\star})$

$\mathcal{C}_{train}, \mathcal{C}_{val} = \mathrm{RandomSplit}(\mathcal{C}, \frac{1}{5})$

$\mathrm{best}_{val} = \infty$

**for** $N_{\mathrm{iter}}$ **do**

$\quad \psi \leftarrow \psi - \alpha \nabla_\psi \left[ \sum_{(\theta, \mathbf{x}_o) \in \mathcal{C}_{train}} |\mathbf{g}_\psi(\mathbf{x}_o) - \mathbb{E}_\varepsilon[\mathbf{h}_{\omega^\star}(S(\theta, \varepsilon))]|_2 \right]$

$\quad \mathrm{cur}_{val} = \sum_{(\theta, \mathbf{x}_o) \in \mathcal{C}_{val}} |\mathbf{g}_\psi(\mathbf{x}_o) - \mathbb{E}_\varepsilon[\mathbf{h}_{\omega^\star}(S(\theta, \varepsilon))]|_2$

$\quad$ **if** $\mathrm{cur}_{val} < \mathrm{best}_{val}$ **then**

$\quad\quad \mathrm{best}_{val} = \mathrm{cur}_{val}$

$\quad\quad \psi^\star = \psi$

$\quad$ **end if**

**end for**

**Step 3: Generate Simulations for Test Set ($N_s = N_o$)**

$\mathcal{S} = \{\mathbf{x}_s^j\}_{j=1}^{N_s}$,

where $\mathbf{x}_s^j \sim S(\theta^j, \varepsilon) \quad \theta^j \sim \pi(\theta) \quad \varepsilon \sim \mathcal{U}[0,1]$

**Step 4: Entropic-regularized OT**

$$C_{ij} = |f_{\omega^\star}(\mathbf{x}_s^j) - g_{\psi^\star}(\mathbf{x}_o^i)| \quad \forall i, j \in \{1, \dots, N_o\} \times \{1, \dots, N_s\}$$

$$P^\star = \arg\min_{P \in \mathcal{B}_o} \langle P, C \rangle + \rho\, KL\left(P^T \mathbf{1}_{N_o} \| \frac{\mathbf{1}_{N_s}}{N_s}\right) + \gamma \langle P, \log P \rangle$$

**Step 5: Compute Posterior Distributions**

$$p(\theta | \mathbf{x}_o^i) := \sum_{j=1}^{N_s} P_{ij}^\star \tilde{p}\left(\theta \mid \mathbf{h}_{\omega^\star}(\mathbf{x}_s^j)\right)$$

**Return** $\tilde{p}(\theta | \mathbf{x}_o^i) \quad \forall \mathbf{x}_o^i \in \mathcal{D}$

---

## C Prior Misspecification Experiments

**Prior misspecification on Task C.** With this experiment we aim to better understand the role of $\tau$ when RoPE is applied with different levels of prior misspecification. We thus re-use the same setup as in Figure 1 but add prior misspecification as a mixture between the assumed prior and a much tighter uniform distribution. As the weight of the tighter uniform distribution increases, the prior gets more misspecified. The experimental setup follows closely the one in the well-specified case (see Section I.2), except calibration samples are drawn from the true prior (as this would be the case in a real-world application) and we compute the OT coupling for values of $\tau \in [0.1, 1]$.

The results in Figure 3b demonstrate that RoPE can be robust to prior misspecification. In particular, we observe that $\tau$ plays the expected role and that values below 1. enable RoPE to perform better when the true prior is only a subset of the prior used to generated synthetic data.

**Prior misspecification on Task E.** In some practical settings, it is unlikely that the prior used to generate synthetic data will match the distribution of the target parameters in the real data. For this reason, we consider a semi-balanced formulation of OT, providing the flexibility to discard simulations with no corresponding real-world observations.

To evaluate the effect of a misspecified prior on RoPE and RoPE⋆, we perform an experiment that would resemble its use in real applications like the ones we outline in the introduction. In such settings—e.g., inferring cardiac parameters or chemical concentrations—the target parameters are limited to a range of validity, and a likely choice for the practitioner would be to select a uniform prior over this range.

To replicate this setting, we collect a new real-world dataset from the light tunnel (Task E) and train RoPE on synthetic data originating from a uniform prior, as we do for the results shown in Figure 1. However, we then apply RoPE to real data generated from a different (betabinomial) distribution over the target parameters.

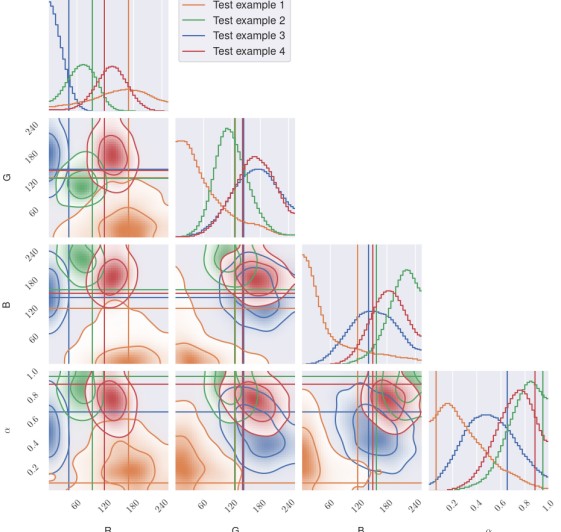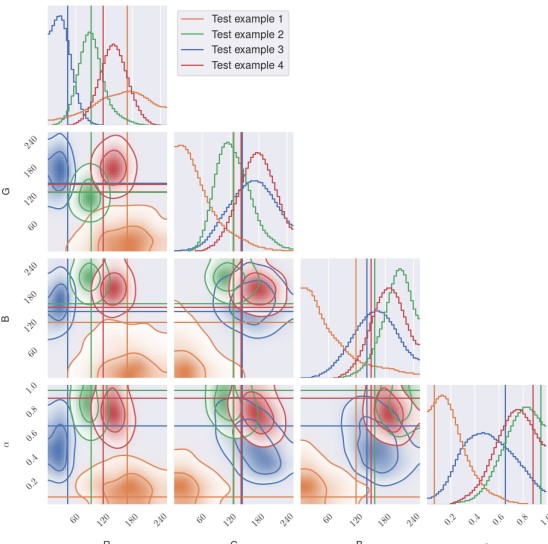

Figure 5: **Visualization of estimated posteriors.** Corner plots of the posteriors estimated by RoPE in the prior-misspecification experiment from Fig. 1 above. We show, in different colors, the estimates for four observations sampled at random from the test set, for RoPE (left) and RoPE⋆ ($\tau = 0.5$) (right) formulation of the OT step, and a calibration set of size 50; the horizontal and vertical lines correspond to the ground-truth value of the parameters.

## D    Robustness to Distribution Shifts

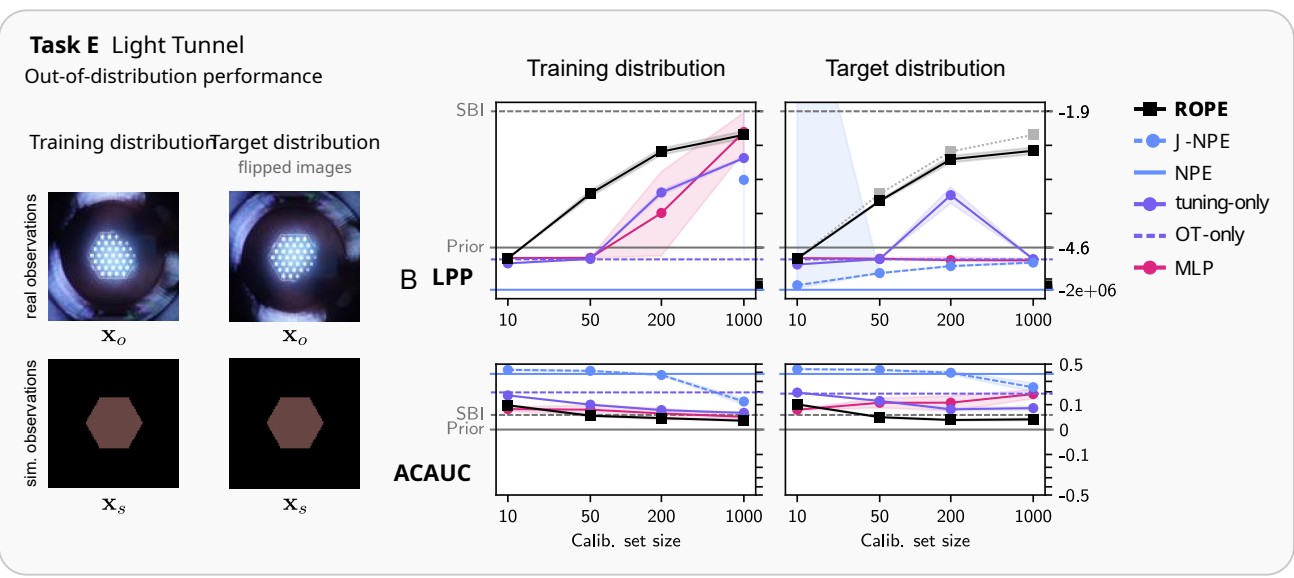

Figure 6: Out-of-distribution performance of RoPE and some baselines. We train RoPE and other baselines on the same light-tunnel data as in task E (training distribution), but apply it to test sets originating from a target distribution where the real-world images are flipped vertically. We compare the performance on test sets from both distributions, showing the LPP and ACAUC scores for each method. For comparison, in the right plot we show again the LPP curve (light gray, dotted) attained by RoPE under the training distribution. The performance of RoPE is barely affected as it cannot exploit any signal in the real images ($\mathbf{x}_o$) beyond what is encoded in the simulator, and the simulator output ($\mathbf{x}_s$) is invariant to the transformation we consider. Because NPE is not trained on real observations, its performance, although poor, also remains virtually unchanged. On the other hand, the performance of MLP and J-NPE drops in the target distribution, as these methods are not limited in what information they can exploit from the real observations on which they are trained, potentially learning shortcuts that are not present in the target distribution. This results demonstrate that if the simulator embeds the right invariances, our modeling assumption $\mathbf{x}_o \perp \theta \mid \mathbf{x}_s$ can be favorable to out-of-distribution generalization.

## E    Optimal Transport Coupling as a joint distribution

With our conditional independence assumption, the problem of modeling $p(\mathbf{x}_o \mid \theta)$ reduces to modeling $p(\mathbf{x}_o \mid \mathbf{x}_s)$ instead. If we assume the prior well-specified, this task is equivalent to modeling $p(\mathbf{x}_o, \mathbf{x}_s)$ under the constraint that the corresponding marginal $p(\mathbf{x}_s) = \int p(\mathbf{x}_s, \mathbf{x}_o)d\mathbf{x}_o$ equals $\int p(\theta)p(\mathbf{x}_s \mid \theta)d\theta$. By construction, the OT coupling, $\pi^\star$, respects the constraint on the marginals, $\int \pi^\star(\mathbf{x}_s, \mathbf{x}_o)d\mathbf{x}_o = p(\mathbf{x}_s)$ and $\int \pi^\star(\mathbf{x}_s, \mathbf{x}_o)d\mathbf{x}_s = p(\mathbf{x}_o)$ , and the exact instantiation $\pi^\star$ depends also on the chosen cost function which can always be defined to yield any given conditional $p(\mathbf{x}_o \mid \mathbf{x}_s)$ that respects the constraint $\int p(\mathbf{x}_o \mid \mathbf{x}_s)p(\mathbf{x}_s)d\mathbf{x}_s = p(\mathbf{x}_o)$. $\pi^*$ can thus model the "right" posterior, provided the right cost function is used. In the case, where the prior cannot be trusted, we suggest to use $\tau < 1$ and relax the OT formulation. In this case, we only enforce that all elements of $p(x_o)$ are matched to a subset of the elements of $p(x_s)$. This implicitly assumes that the assumed prior $p(\theta)$ is overly conservative and covers $p^\star(\theta)$. We believe this is a reasonable assumption as it is often easy to derive physical bounds for the parameter values and use a uniform distribution.

## F  Self-calibration Property

We say RoPE is self-calibrating because, by design, the posterior distribution marginalized over observations tends to the prior as the number of simulation increases, that is,

$$\int_{\mathcal{X}} \tilde{p}(\theta \mid \mathbf{x}_o) p(\mathbf{x}_o) d\mathbf{x}_o = p(\theta). \tag{7}$$

This property is also called marginal calibration, and is a necessary condition for a posterior estimation method to be calibrated. Considering NPE, $\tilde{p}(\theta \mid \mathbf{x}_s)$, is marginally calibrated and observations $\mathbf{x}_o$ are generated from the assumed prior, that is sampled from an unknown distribution $p(\mathbf{x}_o) = \int p(\mathbf{x}_o \mid \theta) p(\theta)$, we can show RoPE is marginally calibrated. Indeed, considering the Monte-Carlo approximation of the marginalized posterior distribution over the test set $\mathcal{D}_o := \{\mathbf{x}_o^i\}_{i=1}^{N_o}$, we have,

$$\int_{\mathcal{X}} \tilde{p}(\theta \mid \mathbf{x}_o) p(\mathbf{x}_o) d\mathbf{x}_o = \mathbb{E}_{p(\mathbf{x}_o)}[\tilde{p}(\theta \mid \mathbf{x}_o)] \tag{8}$$

$$\approx \frac{1}{N_o} \sum_{i=1}^{N_o} \tilde{p}(\theta \mid \mathbf{x}_o^i) \tag{9}$$

$$= \frac{1}{N_o} \sum_{i=1}^{N_o} \sum_{j=1}^{N_s} N_o P_{ij}^{\star} \tilde{p}(\theta \mid \mathbf{x}_s^j) \tag{10}$$

$$= \sum_{j=1}^{N_s} \left[ \sum_{i=1}^{N_o} P_{ij}^{\star} \right] \tilde{p}(\theta \mid \mathbf{x}_s^j) \tag{11}$$

$$= \frac{1}{N_s} \sum_{j=1}^{N_s} \tilde{p}(\theta \mid \mathbf{x}_s^j) \tag{12}$$

$$\approx p(\theta), \tag{13}$$

where we use the definition of the transport matrix to get $\sum_{i=1}^{N_o} P_{ij}^{\star} = \frac{1}{N_s}$. The last approximation tends to be exact as the number of simulations increases, if the NPE is marginally calibrated.

## G  Learning Minimal Sufficient Statistics with Neural Posterior Estimation

We now discuss why NPE may learn a minimal sufficient statistic under perfect training. First, under a sufficiently large validation set, NPE's objective function is only optimal on the validation set if NPE models the true posterior as defined implicitly by the prior $p(\theta)$ and the likelihood corresponding to the simulator $S$. This consistency has been proven in (Papamakarios & Murray, 2016) and is the motivation to use such an objective when estimating density. Second, some normalizing flows, such as autoregressive UMNN flows (Wehenkel & Louppe, 2019), are universal approximators of continuous densities. In addition, neural networks are also universal function approximators. As such, we can claim that it is always possible to parameterize the NCDE $p_\theta(\theta \mid \mathbf{h}_\omega(\mathbf{x}))$ such that the class of functions its parameters represent contains the true posterior. We directly observe that $\mathbf{x}$ is only used by the NCDE through $\mathbf{h}_\omega(\mathbf{x})$. Thus, under perfect training $p_{\theta^\star}(\theta \mid \mathbf{h}_{\omega^\star}(\mathbf{x})) = p(\theta \mid \mathbf{x})$ and $\mathbf{h}_{\omega^\star}(\mathbf{x})$ is a sufficient statistic for $\theta$ given $\mathbf{x}$ under the simulator's model.

Without additional constraints, we cannot claim anything about the minimality of $\mathbf{h}_{\omega^\star}(\mathbf{x})$. Nevertheless, we can enforce the neural network $\mathbf{h}_{\omega^\star}(\mathbf{x})$ to have an information bottleneck and thus reduce the information carried. In practice, we choose the output dimension of $\mathbf{h}_{\omega^\star}(\mathbf{x})$ so that the NCDE achieves optimal performance on the test set. Because in the context of SBI we can generate as many (simulated) samples as needed, we can obtain estimators that closely approach the simulation's posterior and a minimal sufficient statistic.

# H    Computational cost of RoPE

Running NPE is broadly recognized as having a low computational cost: once the upfront training is complete, the cost of inverting the normalizing flow to sample from the posterior during inference becomes negligible as the number of test observations increases. This makes NPE more efficient than methods like Approximate Bayesian Computation or Markov Chain Monte Carlo (when the simulator allows likelihood evaluation). RoPE introduces additional computational costs on top of running NPE: (1) the OT coupling computation, i.e., solving (2), and (2) obtaining samples from the estimated posterior distributions, to compute the posterior estimate defined in (5). The computational cost of solving the transport problem with the Sinkhorn algorithm (Cuturi, 2013) is quadratic in the number of real-world observations. The sampling step has a negligible cost as it directly sub-samples from the set of points generated with NPE.

In our experiments, solving the OT optimization for $2000$ test examples takes less than a minute on an M1 MacBook Pro. Sampling from the mixture of posterior distributions involves caching 10,000 samples for each simulation and generating 5,000 samples by sub-sampling from the mixture using the OT coupling matrix. This caching process takes under three minutes, and is comparable to the cost of running NPE alone.

Extending RoPE to handle larger test sets or an online setting (processing test examples one at a time) is outside the scope of this work. Nevertheless, mehtods like Neural OT (e.g., (Makkuva et al., 2020)) and online Sinkhorn (Mensch & Peyré, 2020) should provide good solutions to make RoPE fully amortized.

# I    Experimental Setup

In this section, we provide more details on our experiments. For completeness, we provide details on the neural architectures and training hyperparameters. However, we encourage the reader interested in reproducing our experiments to examine our code directly (a link to the code will be made available in the public version of the paper).

For all methods training on calibration set we keep always keep $20\%$ of the calibration to monitor validation performance and we select the best model based on this metric.

For the MLP we use the same architecture as the NSE for all our experiments and optimize its parameters on the calibration set with Adam and a learning rate equal to $0.0003$, we select the best model based on the LPP attributed to the validation subset of the calibration set.

**Computing the SBI baseline.** We take the ground-truth labels $\{(\theta^i\}_{i=1}^N$ from the test set $\{\theta^i, \mathbf{x}_o^i)\}_{i=1}^N$ on which we compute all the metrics for Figure 1; for each label $\theta^i$, we simulate a synthetic observation $\mathbf{x}_s^i := \mathcal{S}(\theta^i)$, collecting them into a "synthetic" test set $\{(\theta^i, \mathbf{x}_s^i)\}_{i=1}^N$; then, we apply to it the NSE+NPE pipeline (simulated posterior in Figure 4, right) to obtain the posterior estimates which we then evaluate. In this way, the baseline represents the performance we would hope to achieve if there was no misspecification and the simulator perfectly replicated the real observations (up to the stochasticity of the simulator itself).

## I.1    Task A: CS & Task B: SIR

**Task A (synthetic): CS.** We reproduce the cancer and stromal cell development benchmark from Ward et al. (2022). The simulator emulates the development of cancer and stromal cells in a 2D environment as a function of three Poisson rate parameters $(\lambda_c, \lambda_p, \lambda_d)$. The observations are vectors composed of the number of cancer and stromal cells and the mean and maximum distance between stromal cells and their nearest cancer cell. Synthetic misspecification is introduced by removing cancer cells that are too close to their generating parent.

**Task B (synthetic): SIR.** We also use the stochastic epidemic model from Ward et al. (2022), which describes epidemic dynamics through the infection rate $\beta$ and recovery rate $\gamma$. Each observation is a vector composed of the mean, median, and maximum number of infections, the day of occurrence of the maximum number of infections, the day at which half the total number of infections was reached, and the mean auto-correlation (lag 1) of the infections. Misspecification is a delay in weekend infection counts, of which $5\%$ are added to the count of the following Monday.

We refer the reader to Ward et al. (2022) for more details about the simulator and prior distribution. We use the exact same setting as theirs.

NEURAL ARCHITECTURE & TRAINING HYPERPARAMETERS

For all methods we use the same backbone MLP as the NSE with ReLU activations and layers composed of $[4K, 16K, 16K, 12K, 3K]$ neurons, where $K$ is the dimensionality of $\theta$. The NF is a 1-step UMNN-MAF (Wehenkel & Louppe, 2019) with $[100, 100, 100]$ neurons for both the autoregressive conditioner and normalizer. For NNPE, we train the UMNN-MAF on simulations poluted by Spike and Slab errors. We train models with Adam and a learning rate equal to $0.0005$ and all other parameters set to default. We optimize the SBI model for $10^6$ gradient steps and select the best model on random validation sets containing $10^5$ simulations.

## I.2 Task C: Pendulum

DESCRIPTION

The first task is inspired from the damped pendulum benchmark commonly used to assess hybrid learning algorithms. Given a 2D physical parameter $\theta := [\omega_0, A]$, where $\omega_0 \in \mathbb{R}^+$ denotes the fundamental frequency and $A \in \mathbb{R}^+$ the amplitude of a friction-less pendulum, the simulator generates the horizontal position of the pendulum at $200$ discrete times during uniformly sampled in a $10$ seconds interval as

$$\mathbf{x}_s := [\theta(t = 0), \dots, \theta(t = 10s)] \in \mathbb{R}^{200}$$
$$\text{where } \theta(t) = A\cos(\omega_0 t + \varphi) \quad \varphi \sim \mathbb{U}(-\pi, \pi). \tag{14}$$

The relationship between the parameters and the simulation is thus stochastic as $\varphi$ accounts for an unknown phase shift when the measurements start. We generate real-world observations synthetically by replacing $\theta(t)$ from (14) by

$$\tilde{\theta}(t) = e^{\alpha t} A\cos(\omega_0 t + \varphi) \quad \varphi \sim \mathbb{U}[-\pi, \pi] \quad \alpha \sim \mathbb{U}[0, 1],$$

where $\alpha$ represents the effect of friction. We also add Gaussian noise on both simulated and real-world data to represent the inaccuracy of a sensor measuring the pendulum's position. The prior distribution is a product of uniform distribution, $p(\theta := [\omega_0, A]) = \mathcal{U}[0, 3] \times \mathcal{U}[0.5, 10]$.

NEURAL ARCHITECTURE & TRAINING HYPERPARAMETERS

**Neural Posterior Estimator.** The NSE is a 1D convolutional neural network, with the architecture described in Algorithm 2. The NCDE is a one-step discrete normalizing flow with an autoregressive conditioner and a UMNN (Wehenkel & Louppe,

---

**Algorithm 2** Convolutional Neural Network for Tasks A and D.

---

1: $\mathrm{Conv1d}(1, 16, 3, 1, \mathrm{dilation} = 2, \mathrm{padding} = 1)$
2: $\mathrm{ReLU}()$
3: $\mathrm{Conv1d}(16, 64, 3, 2, \mathrm{dilation} = 2, \mathrm{padding} = 1)$
4: $\mathrm{ReLU}()$
5: $\mathrm{AvgPool1d}(3, 1)$
6: $\mathrm{Conv1d}(64, 128, 3, 1, \mathrm{dilation} = 2, \mathrm{padding} = 1)$
7: $\mathrm{ReLU}()$
8: $\mathrm{Conv1d}(128, 128, 3, 2, \mathrm{dilation} = 2, \mathrm{padding} = 1)$
9: $\mathrm{ReLU}()$
10: $\mathrm{AvgPool1d}(3, 1)$
11: $\mathrm{Conv1d}(128, 128, 3, 1, \mathrm{dilation} = 2, \mathrm{padding} = 1)$
12: $\mathrm{ReLU}()$
13: $\mathrm{Conv1d}(128, 128, 3, 2, \mathrm{dilation} = 2, \mathrm{padding} = 1)$
14: $\mathrm{ReLU}()$
15: $\mathrm{AvgPool1d}(3, 1)$
16: $\mathrm{Conv1d}(128, 128, 3, 1, \mathrm{dilation} = 2, \mathrm{padding} = 1)$
17: $\mathrm{ReLU}()$
18: $\mathrm{Flatten}()$
19: $\mathrm{Linear}(2048, 512)$
20: $\mathrm{ReLU}()$
21: $\mathrm{Linear}(512, 128)$
22: $\mathrm{ReLU}()$
23: $\mathrm{Linear}(128, 32)$
24: $\mathrm{ReLU}()$
25: $\mathrm{Linear}(32, 10)$

---

2019) as the normalizer. The autoregressive conditioner is a MADE with ReLU activation and 3 layers of 100 neurons that output a 10 dimensional vector to the UMNN. The UMNN has an integrand net with 3 layers of 100 neurons with ReLU activations. For training the NPE, we use a batch size of 100 and a learning factor equal to 1e-4. NPE is trained until convergence. Other parameters are set to default values and should marginally impact the NPE obtained.

**Algorithm 3** UNet1D Architecture

1: Unet1D :
2:     Encoder1D :
3:         Block(in_channels $= 1$, out_channels $= 64$)
4:         Block(in_channels $= 64$, out_channels $= 128$)
5:         Block(in_channels $= 128$, out_channels $= 256$)
6:         Block(in_channels $= 256$, out_channels $= 512$)
7:         Block(in_channels $= 512$, out_channels $= 1024$)
8:         MaxPool1d(2)
9:     Decoder1D :
10:         ConvTranspose1d($1024 + 5, 512, 2$, stride $= 2$)
11:         Block(in_channels $= 1024$, out_channels $= 512$)
12:         ConvTranspose1d($512, 256, 2$, stride $= 2$)
13:         Block(in_channels $= 512$, out_channels $= 256$)
14:         ConvTranspose1d($256, 128, 2$, stride $= 2$)
15:         Block(in_channels $= 256$, out_channels $= 128$)
16:         ConvTranspose1d($128, 64, 2$, stride $= 2$)
17:         Block(in_channels $= 128$, out_channels $= 64$)
18:         ConvTranspose1d($64, 1, 2$, stride $= 2$)
19:         Block(in_channels $= 64$, out_channels $= 1$)
20:         Conv1d($64, 1, 1$)

**Algorithm 4** Block1D(in_channels, out_channels)

1: Conv1d(in_channels, out_channels, kernel_size=3, padding=1)
2: ReLU()
3: Conv1d(out_channels, out_channels, kernel_size=3, padding=1)
4: ReLU()

**Algorithm 5** 2D Convolutional Neural Network

1: Conv2d(3, 64, 3, 2, dilation=1), ReLU()
2: Conv2d(64, 128, 3, 2, dilation=1), ReLU()
3: MaxPool2d(3)
4: Conv2d(128, 128, 3, 2, dilation=1), ReLU()
5: Conv2d(128, 64, 1, 1, dilation=1), ReLU()
6: Conv2d(64, 3, 1, 1, dilation=1), ReLU()
7: Flatten()
8: Linear(27, 100), ReLU()
9: Linear(100, 20)

**RoPE NSE.** We have selected the best NPE based on the validation set with $10000$ examples generated with the simulator. The NPE is fixed to one best-of-all model. We fine-tune the NCDE with a learning rate equal to 1e-5 for $5000$ gradient steps on $80\%$ the full calibration set. We use a 1-sample Monte Carlo estimate of the expectation in (6).

**J-NPE.** To train J-NPE, we simply randomly use a batch composed of $50\%$ of simulated pairs $(\theta, \mathbf{x}_s)$ and of $50\%$ $(\theta, \mathbf{x}_o)$ from the calibration set. We use the same architecture and hyper-parameters as the SBI NPE. The best model is selected based on the best training set performance. We do $50$ epochs with $50000$ simulated examples for each epoch. The batch size is $100$.

**HVAE.** For the HVAE, we re-use the NPE model as the physics encoder and replace the decoder with a deterministic version of the simulator, thus removing the Gaussian noise on a random phase shift. In addition, we follow the approach of Takeishi & Kalousis (2021) and have 1) a real-world encoder that maps $\mathbf{x}_o$ to $\mathbf{z}_a$, 2) a reality-to-physics encoder, and 3) a physics-to-reality decoder. The real-world encoder has the same architecture as the NSE of the NPE and outputs the mean and log-variance of a 5D latent vector $\mathbf{z}_a$. The reality-to-physics and physics-to-reality also have the same architectures and are two conditional 1D U-Net with neural network architecture described in Algorithm 3.

To train the HVAE, we freeze the parameters of the NPE and optimizes the ELBO as well as a calibration loss that evaluates the likelihood assigned to the true physical parameters. All distributions are parameterized by Gaussian with mean and log-variance predicted by the neural networks. We do not use any additional losses as we expect constraining NPE and using the calibration set should already provide the necessary support to use the physics in a meaningful way. The HVAE is trained on the $2000$ test examples as it is the only real-world data, calibration set aside, that we have access to. We use a batch size equal to $100$ and a learning rate equal to 1e-3. We believe obtaining a better HVAE is possible. However, we emphasize the complexity of setting up a good HVAE for the only purpose of statistical inference over parameters.

DATASETS

For this task, we can generate samples $(\theta, \mathbf{x}_s)$ on the fly to train the NPE. The calibration and test sets are also generated randomly by sampling from the prior distribution and using the damped pendulum simulator.

## I.3 Task D: Hemodynamics

### DESCRIPTION

Inspired by Wehenkel et al. (2023), we define the task of inferring important cardiovascular parameters from normalized arterial pressure waveforms measured at the radial artery. The simulator uses many physiological parameters that modulates the heart function, physical properties of the $116$ main arterial segments, and behavior of the vascular beds. Our inference concerns two parameters of the heart function, $\theta := [\text{SV}, \text{LVET}]$, the stroke volume (SV) is the amount pumped out from the left ventricle over the heart beat modeled, and the left ventricular ejection time (LVET) is the time interval between opening and closure of the aortic valve. Other parameters, such as the heart rate or arteries' stiffness, are considered as nuisance effects and are randomly sampled from a realistic population distribution. An additional source of randomness is added by modeling measurement errors with a white Gaussian noise and randomizing the starting recording time with respect to the cardiac cycle. The simulator produces $8$-second timeseries $\mathbf{x}_t \in \mathbb{R}^{1000}$ sampled at $125Hz$. As synthetic misspecification, the simulator assumes all arteries have the same length over the population considered, whereas "real-world" data are artificially generated by also varying the length of arteries and account for the effect of human's height. The simulator is based on the openBF PDE solver (Melis, 2017) specialized for hemodynamics, which is not differentiable and takes approximately one minute to simulate one sample on a standard CPU. This synthetic tasks represent a common scenario in which a simulator, although faithful to the effect of certain parameters, misses additional degrees of freedom that exists for the real-world data.

### NEURAL ARCHITECTURE & TRAINING HYPERPARAMETERS

---

**Algorithm 6** CNN Architecture for Task C.

---

1: Conv1d(1, 16, 3, 1, dilation=2, padding=1), ReLU()
2: Conv1d(16, 64, 3, 2, dilation=2, padding=1), ReLU()
3: AvgPool1d(4, 2)
4: Conv1d(64, 128, 3, 1, dilation=2, padding=1), ReLU()
5: Conv1d(128, 128, 3, 2, dilation=2, padding=1), ReLU()
6: AvgPool1d(4, 2)
7: Conv1d(128, 128, 3, 1, dilation=2, padding=1), ReLU()
8: Conv1d(128, 128, 3, 2, dilation=2, padding=1), ReLU()
9: AvgPool1d(4, 1)
10: Conv1d(128, 128, 3, 1, dilation=2, padding=1), ReLU()
11: Flatten()
12: Linear(1024, 512), ReLU()
13: Linear(512, 128), ReLU()
14: Linear(128, 32), ReLU()
15: Linear(32, 5)

---

**Neural Posterior Estimator.** The NSE is the 1D convolutional neural network described in Algorithm 6. The NCDE is a $5$-step discrete normalizing flow with an autoregressive conditioner and affine normalizers. Each of the $5$ autoregressive conditioners is a MADE with ReLU activations and $4$ layers of $300$ neurons that output $4$ dimensional vectors used to parameterize the affine transformations. For training the NPE, we use a batch size of $100$ and a learning factor equal to 5e-4. NPE is trained until convergence. Other parameters are set to default values and should marginally impact the NPE obtained.

**RoPE NSE.** We have selected the best NPE based on the validation set with $2000$ examples generated with the simulator. The NPE is fixed to one best-of-all model. We fine-tune the NCDE with a learning rate equal to 1e-5 for $2000$ gradient steps on $80\%$ of calibration set. We use a 1-sample Monte Carlo estimate of the expectation in (6).

**J-NPE.** To train J-NPE, we simply randomly use a batch composed of $50\%$ of simulated pairs $(\theta, \mathbf{x}_s)$ and of $50\%$ $(\theta, \mathbf{x}_o)$ from the calibration set. We use the same architecture and hyper-parameters as the SBI NPE. The best model is selected based on the best training set performance. We do $50$ epochs with $6000$ simulated examples for each epoch. The batch size is $100$.

**HVAE.** There is no HVAE for this experiment as the simulator is non-differentiable.

For this task, we cannot generate samples $(\theta, \mathbf{x}_s)$ on the fly to train the NPE. For the purpose of this experiment, we have generated $10000$ simulations and real-world observations. Our fine-tuning strategy approximates (6) by finding the simulations with the closest parameter value.

## I.4   Task E: Light Tunnel

DESCRIPTION

We use one of the light-tunnel datasets from the causal chamber project (Gamella et al., 2025, `causalchamber.org`). In particular, we use the data from the `ap_1.8_iso_500.0_ss_0.005` experiment in the `lt_camera_v1` dataset. The light tunnel is an elongated chamber with a controllable light source at one end, two linear polarizers mounted on rotating frames, and a camera that takes images of the light source through the polarizers. We refer the reader to Gamella et al. (2025, Figure 2) for a complete schematic. Our task consists of predicting the color setting of the light source $((R, G, B) \in [0, 255]^3)$ and the dimming effect of the linear polarizers $\alpha \in [0, 1]$ from the captured images. As a misspecified simulator of the image-generating process, we adopt the simple model described in Gamella et al. (2025, Model F1, Appendix D). A Python implementation is available through the `causalchamber` package (`models.model_f1`); visit `causalchamber.org` for more details. As input, the simulator takes the parameters $\theta := [R, G, B, \alpha]$ and produces an image consisting of a hexagon roughly the size of the light source, with an RGB color vector equal to $[\alpha R, \alpha G, \alpha B]$. The factor $\alpha := \cos^2(\theta_1 - \theta_2)$, where $\theta_1, \theta_2$ denote the angles of the two polarizers, corresponds to Malus' law (e.g. , Collett, 2005), which models the dimming effect of the polarizers as a function of their relative angle. Besides the obvious misspecification with respect to image realism (see Figure 1), the model ignores other important physical aspects, such as the spectral response of the camera sensor or the non-uniform effect of the polarizers on the different colors—more details can be found in Gamella et al. (2025, Appendix D.IV.2.2). The prior is uniform over colors and polarizer angles, which leads to a non-uniform prior over the dimming effect $\alpha$.

NEURAL ARCHITECTURE & TRAINING HYPERPARAMETERS

**Neural Posterior Estimator.** The NSE is the 2D convolutional neural network described by Algorithm 5.

The NCDE is also a one-step discrete normalizing flow with an autoregressive conditioner and a UMNN (Wehenkel & Louppe, 2019) as the normalizer. The autoregressive conditioner is a MADE with ReLU activation and $3$ layers of $500$ neurons that outputs a $10$ dimensional vector to the UMNN. The UMNN has an integrand net with $4$ layers of $150$ neurons with ReLU activations. For training the NPE, we use a batch size of $100$ and a learning factor equal to 5e-4. NPE is trained until convergence. Other parameters are set to default values and should marginally impact the NPE obtained.

**RoPE NSE.** We have selected the best NPE based on the validation set with $10000$ examples generated with the simulator. The NPE is fixed to one best-of-all model. We fine-tune the NCDE with a learning rate equal to 1e-4 for $2000$ gradient steps on on $80\%$ of the calibration set. We use a 1-sample Monte Carlo estimate of the expectation in (6).

**J-NPE.** To train J-NPE, we simply randomly use a batch composed of $50\%$ of simulated pairs $(\theta, \mathbf{x}_s)$ and of $50\%$ $(\theta, \mathbf{x}_o)$ from the calibration set. We use the same architecture and hyper-parameters as the SBI NPE. The best model is selected based on the best training set performance. We do $50$ epochs with $1000$ simulated examples for each epoch. Simulations are generated randomly for each batch by sampling the prior and simulating for the corresponding parameters. The batch size is $100$.

**HVAE.** For the HVAE, we re-use the NPE model as the physics encoder and use the simulator as is as it is differentiable without additional effort. In addition, we follow the approach of Takeishi & Kalousis (2021) and have 1) a real-world encoder that maps $\mathbf{x}_o$ to $\mathbf{z}_a$, 2) a reality-to-physics encoder, and 3) a physics-to-reality decoder. The real-world encoder has the same architecture as the NSE of the NPE and outputs the mean and log-variance of a $5D$ latent vector $\mathbf{z}_a$. The reality-to-physics and physics-to-reality also have the same architectures and are two conditional 2D U-Net with the architecture described by Algorithm 7.

To train the HVAE, we freeze the parameters of the NPE and optimizes the ELBO as well as a calibration loss that evaluates the likelihood assigned to the true physical parameters. All distributions are parameterized by Gaussian with mean and log-variance predicted by the neural networks. We do not use any additional losses as we expect constraining NPE and using the calibration set should already provide the necessary support to use the physics in a meaningful way. The HVAE is

**Algorithm 7** 2D UNet

```
 1: Encoder2D:
 2:     Block2D(in_channels=3, out_channels=64)
 3:     Block2D(in_channels=64, out_channels=128)
 4:     Block2D(in_channels=128, out_channels=256)
 5:     Block2D(in_channels=256, out_channels=512)
 6:     Block2D(in_channels=512, out_channels=1024)
 7:     MaxPool2d(2)
 8: Decoder2D:
 9:     ConvTranspose2d(1024 + 5, 512, 2, stride=2)
10:     Block2D(in_channels=1024, out_channels=512)
11:     ConvTranspose2d(512, 256, 2, stride=2)
12:     Block2D(in_channels=512, out_channels=256)
13:     ConvTranspose2d(256, 128, 2, stride=2)
14:     Block2D(in_channels=256, out_channels=128)
15:     ConvTranspose2d(128, 64, 2, stride=2)
16:     Block2D(in_channels=128, out_channels=64)
17:     ConvTranspose2d(64, 1, 2, stride=2)
18:     Block2D(in_channels=64, out_channels=1)
19:     Conv2d(64, 1, 1)
```

**Algorithm 8** Block2D(in_channels, out_channels)

```
 1: Conv2d(in_channels, out_channels, kernel_size=3,
         padding=1, bias=False)
 2: BatchNorm2d(num_features=out_channels)
 3: ReLU(inplace=True)
 4: Conv2d(out_channels,        out_channels,
         kernel_size=3, padding=1, bias=False)
 5: BatchNorm2d(num_features=out_channels)
 6: ReLU(inplace=True)
```

trained on the $2000$ test examples as it is the only real-world data, calibration set aside, that we have access to. We use a batch size equal to $100$ and a learning rate equal to 1e-3. We believe obtaining a better HVAE is possible. However, we emphasize the complexity of setting up a good HVAE for the only purpose of statistical inference over parameters.

DATASETS

For this task, we can generate samples $(\theta, \mathbf{x}_s)$ on the fly to train the NPE. However, the calibration and test sets are real-world data. We ensure there is not overlap between calibration and test set. The is no randomization and the test set is constant for all experiments, the calibration set are also fixed for a given calibration set size.

### I.5   Task F: Wind Tunnel

DESCRIPTION

We use one of the wind-tunnel datasets from the causal chamber project (Gamella et al., 2025, `causalchamber.org`). In particular, we use the data from the `load_out_0.5_osr_downwind_4` experiment in the `wt_intake_impulse_v1` dataset. The tunnel is a chamber with two controllable fans that push air through it and barometers that measure air pressure at different locations. A hatch precisely controls the area of an additional opening to the outside (see Gamella et al., 2025, Figure 2). The data is a collection of pressure curves that result from applying a short impulse to the intake fan load and measuring the change in air pressure using one of the barometers inside the tunnel. Our inference task consists of predicting the hatch position, $\theta := [H] \in [0, 45]$ given a pressure curve (see Figure 1). As a simulator model, we combine the models A2 and C3 described in Gamella et al. (2025, Appendix D); we numerically solve the ODE in model A2, and add stochastic components to simulate the sensor noise and the unknown time point at which the impulse is applied. This results in the simulator being neither differentiable nor deterministic. A Python implementation of the complete simulator is available in the `causalchamber` package (`models.simulator_a2_c3`); visit `causalchamber.org` for more details. Misspecification arises from the many simplifying assumptions needed to model the complex dynamics of the airflow inside the tunnel—more details can be found in Gamella et al. (2025, Appendix D.IV.1.2).

**Neural Posterior Estimator.** The NSE and NCDE have the same 1D convolutional neural network as for Task A. For training the NPE, we use a batch size of $100$ and a learning factor equal to 5e-4. NPE is trained until convergence. Other parameters are set to default values and should marginally impact the NPE obtained.

**RoPE NSE.** We have selected the best NPE based on the validation set with $10000$ examples generated with the simulator. The NPE is fixed to one best-of-all model. We fine-tune the NCDE with a learning rate equal to 1e-4 for $20000$ gradient steps on on $80\%$ of the calibration set. We use a 1-sample Monte Carlo estimate of the expectation in (6).

**J-NPE.** To train J-NPE, we simply randomly use a batch composed of $50\%$ of simulated pairs $(\theta, \mathbf{x}_s)$ and of $50\%$ $(\theta, \mathbf{x}_o)$ from the calibration set. We use the same architecture and hyper-parameters as the SBI NPE. The best model is selected based on the best training set performance. We do $50$ epochs with $10000$ simulated examples for each epoch. The batch size is $100$.

**HVAE.** There is no HVAE for this experiment as the simulator is non-differentiable.

DATASETS

For this task, although slightly slower than Task A and B, we can generate samples $(\theta, \mathbf{x}_s)$ on the fly to train the NPE. However, the calibration and test sets are real-world data. We ensure no overlap between the two sets for all calibration set sizes. All sets are fixed for all experiments.

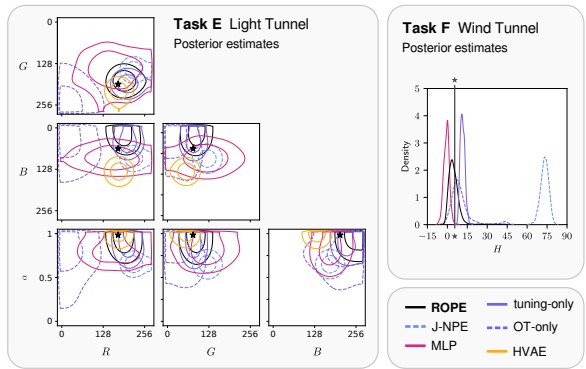

Figure 7: Credible intervals of the posterior estimates at levels $65\%$ and $90\%$, for a single test sample from the light-tunnel task. The black stars denote the true value of the parameter. (center) Posterior estimates for a single test sample from the wind-tunnel task, where the true parameter is denoted by a vertical black line.

## J   Computing ACAUC

---
**Algorithm 9** Statistical Calibration of Posterior Distribution
---
**Input:** Dataset of pairs $\mathcal{D} = \{(\theta^i, \mathbf{x}^i)\}$, Posterior estimator $\tilde{p}(\theta \mid \mathbf{x})$, Number of samples $N$.
**Output:** ACAUC
 1: AVG_CALIBRATION $= 0$
 2: **for** $k \in \{1, \ldots, K\}$) **do**
 3:    Initialize an empty list CredLevels
 4:    **for** $(\theta^i, \mathbf{x}^i) \in \mathcal{D}$ **do**
 5:      Initialize an empty list Samples
 6:      **for** $j = 1$ to $M$ **do**
 7:        Sample $\theta^j$ from $\tilde{p}(\theta \mid \mathbf{x}^i)$
 8:        Append $\theta^j$ to Samples
 9:      **end for**
10:      Sort Samples
11:      Compute the rank (position in ascending order) $r$ of $\theta$ in Samples
12:      Set CredLevels $= \frac{r}{N}$
13:      Append CredLevel to CredLevels
14:    **end for**
15:    Sort CredLevels
16:    CALIBRATION $= \sum_{i=1}^{N}$ CredLevels$[i] - \frac{i}{N}$
17:    AVG_CALIBRATION $=$ AVG_CALIBRATION $+ \frac{\text{CALIBRATION}}{K}$
18: **end for**
**Return:** AVG_CALIBRATION
---

## K   Additional Results

### K.1   Corner plots

### K.2   Calibration plots

## L   Non-iid Calibration Sets

We provide additional results reflecting the behavior of RoPE when the calibration set is not sampled from the "true" prior distribution, on the light tunnel task, when the calibration set comes from a subset of the true distribution. We use the beta distribution of Figure 3b as the calibration set distribution. Figure 15 reports the main metrics (ACAUC and LPP). We observe that, even in this extreme case, RoPE achieves performance that outperforms the prior distribution on the LPP while still being calibrated for calibration set size that are greater than 10. Figure 16 studies how good/bad estimated posteriors are as a function of whether similar samples belongs to the calibration set. As expected, RoPE performs strongly for samples

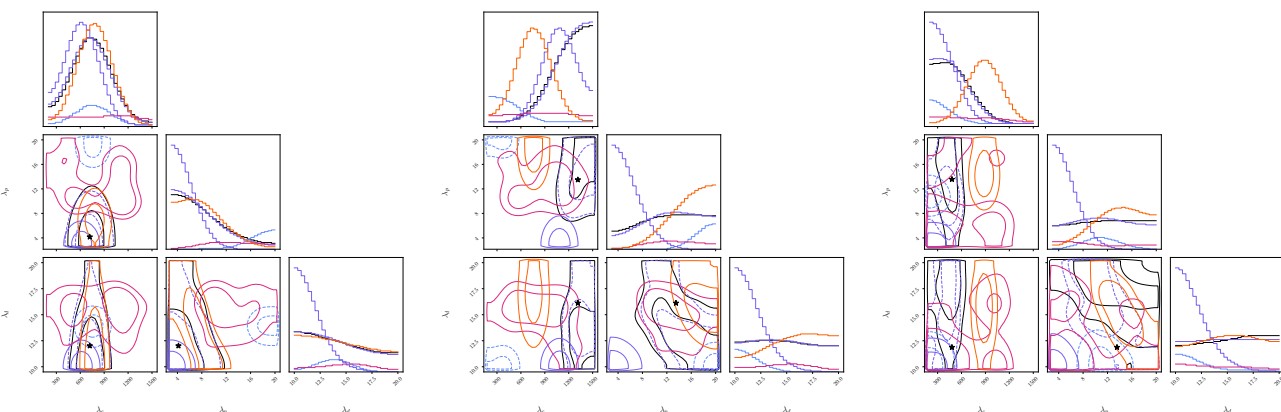

Figure 8: Three corner plots for task A with a calibration set with 50 samples.

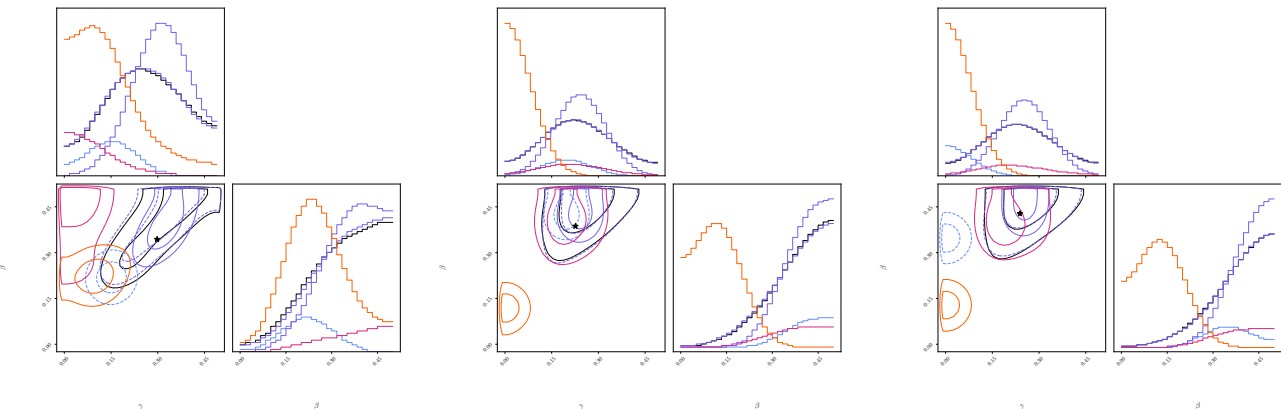

Figure 9: Three corner plots for task B with a calibration set with 50 samples.

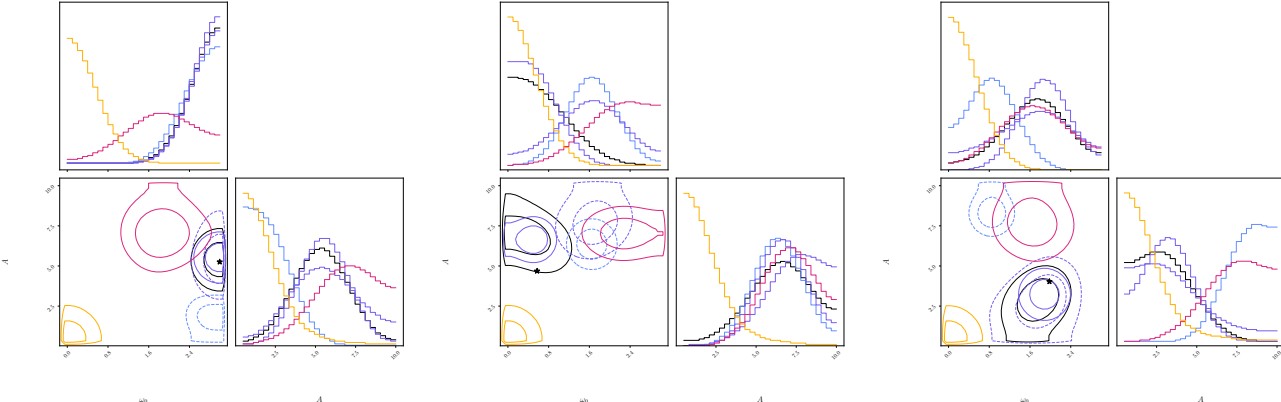

Figure 10: Three corner plots for task C with a calibration set with 50 samples.

that belong to the calibration set while it struggles to generalize to sample that are OOD. Finally, we also show corner plots of the learned posterior in Figure 17 for both samples that are (a) unlikely under the calibration set distribution, and (b) likely under that distribution.

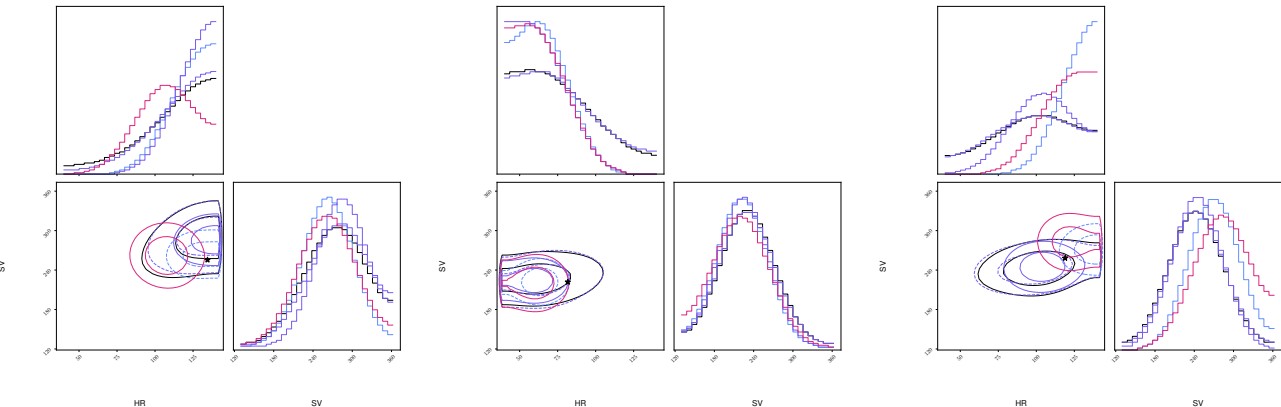

Figure 11: Three corner plots for task D with a calibration set with 50 samples.

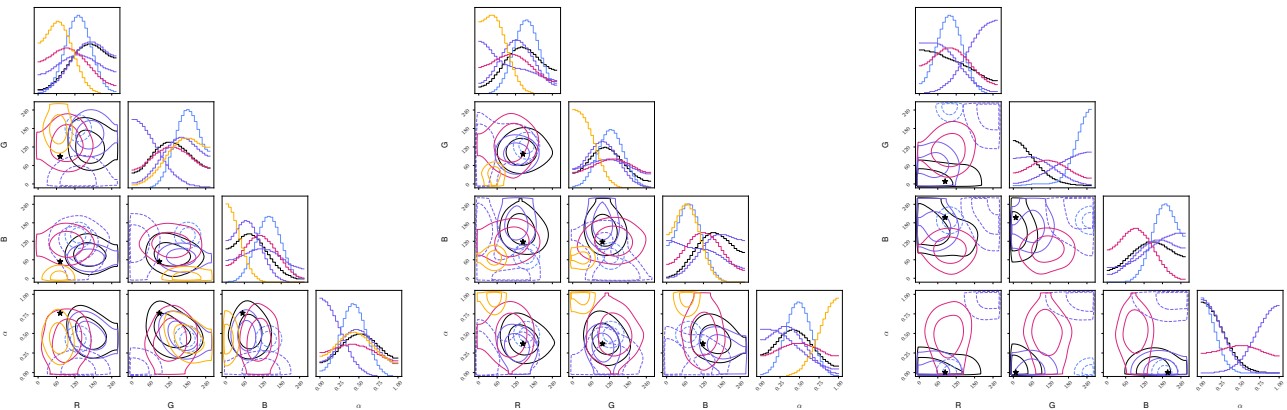

Figure 12: Three corner plots for task E with a calibration set with 50 samples.

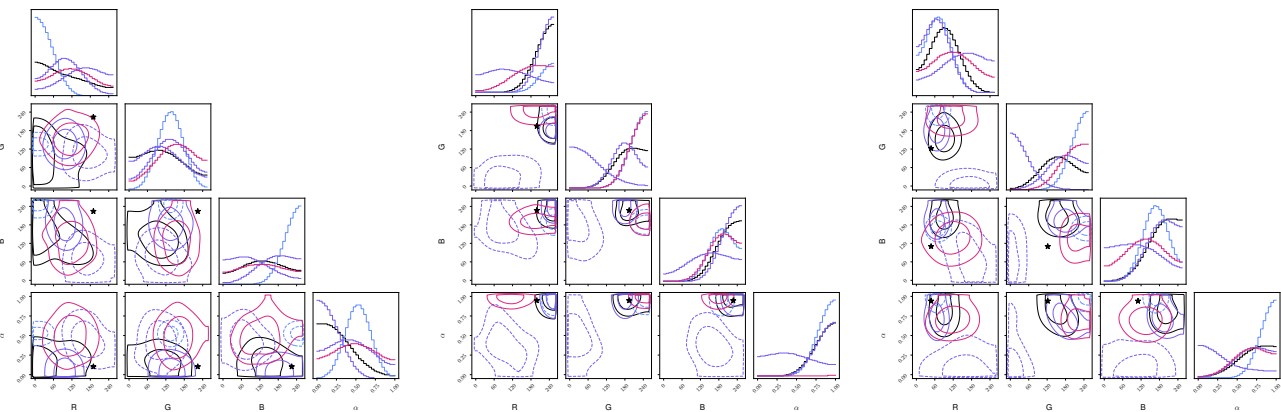

Figure 13: Three corner plots for task E with distribution shift with a calibration set with 50 samples.

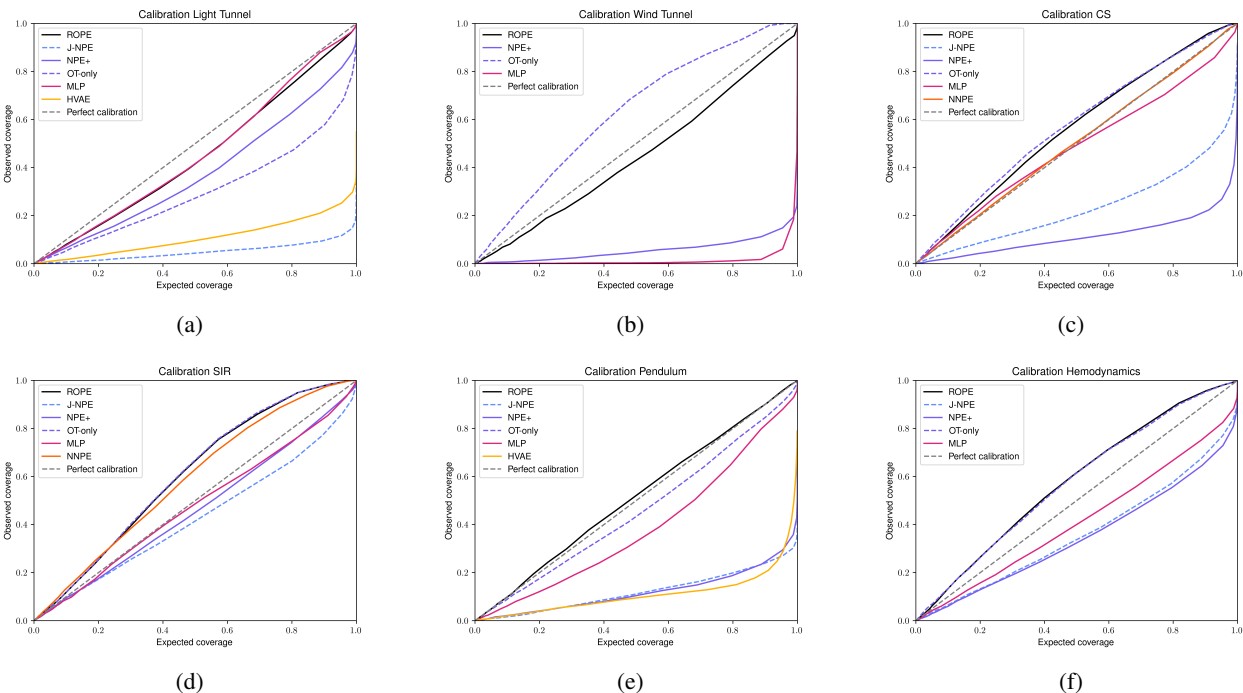

Figure 14: Calibration plots of the different methods on the 6 benchmarks, the coverage at each level is the average of the coverage of the marginal distributions. Each color indicates a different algorithm and the opacity is proportional to the size of the calibration set which ranges from 10 to 1000. We observe that RoPE and OT-only are consistently well calibrated for.

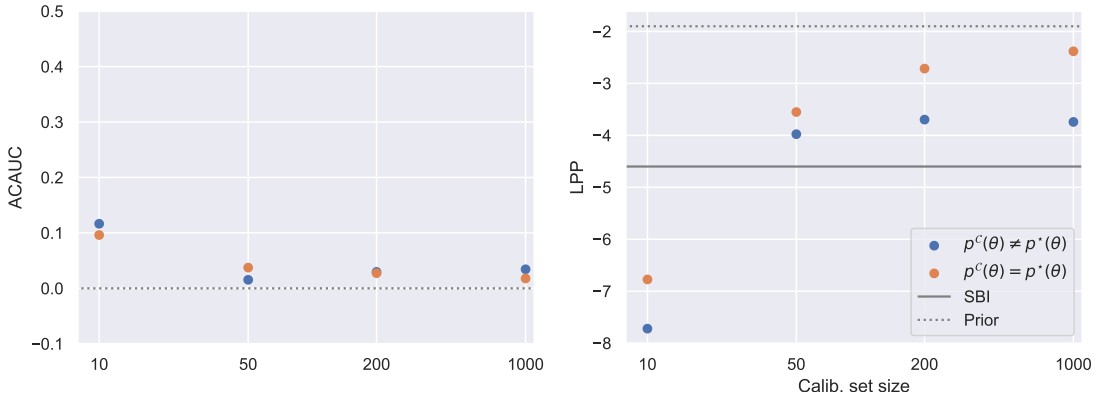

Figure 15: Comparison of ACAUC and LPP for calibration set that is different from the test distribution.

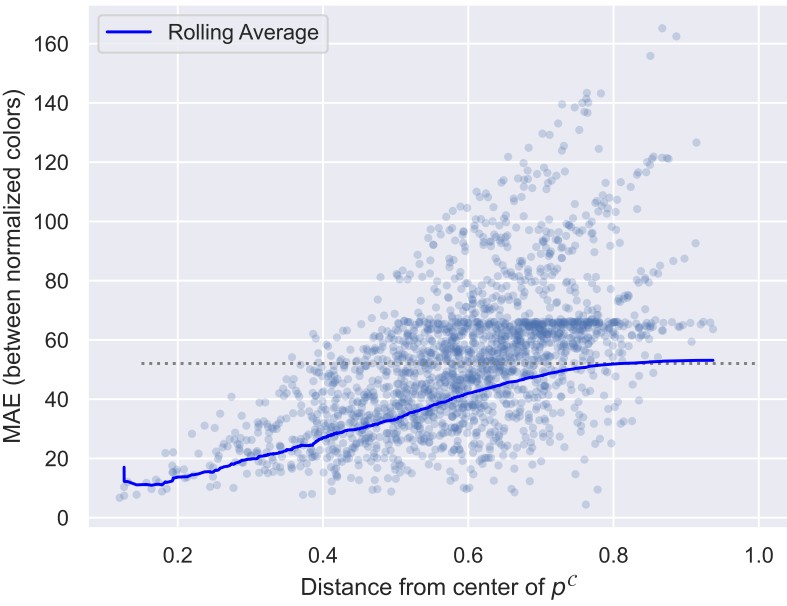

Figure 16: Accuracy of the predicted RGB values (normalized by alpha) as a function of the distance between the analyzed sample from the center of the calibration set distribution.

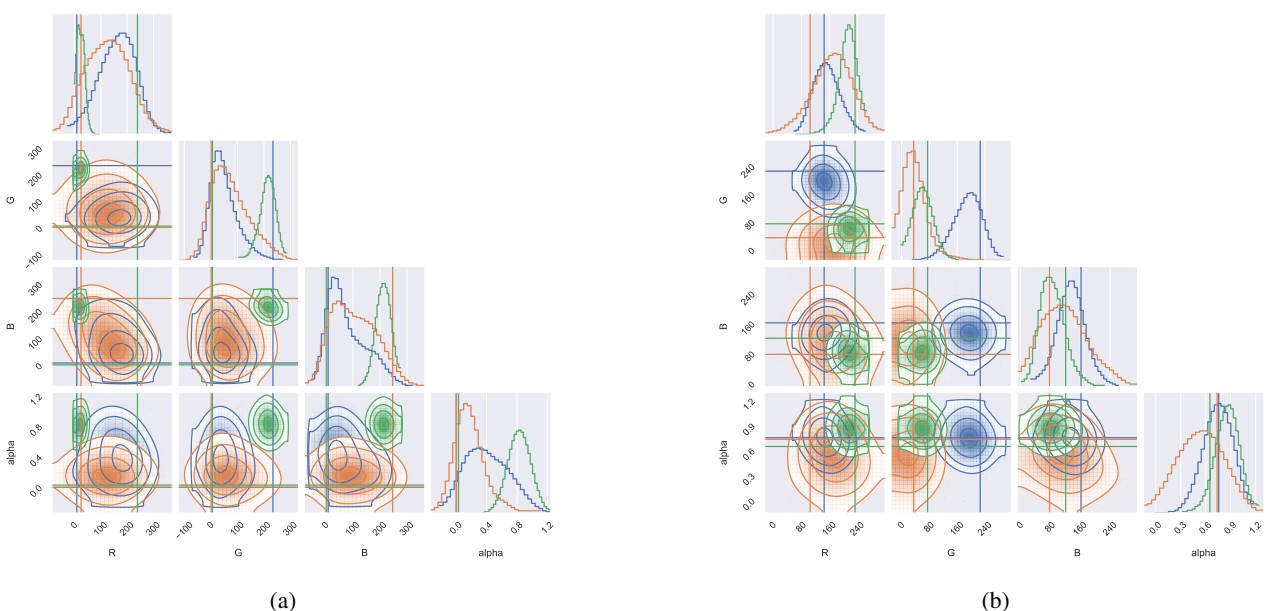

(a)                                                          (b)

Figure 17: corner plots for three distinct observations that are very (a) that are very unlikely under the "bad" calibration set. (b) likely under the "bad" calibration set.

