# OpenReview forum: "Addressing Misspecification in Simulation-based Inference through Data-driven Calibration"
_ICML.cc/2025/Conference — ICML 2025 oral_

### Official Review · Reviewer_6GpC · 2025-03-06

**Overall Recommendation:** 4

**Summary:**

The paper details with the important issue of making neural posterior estimation methods more robust against model misspecification. The authors suggest to use a small set of labeled real-world data to calibrate the posterior inference in the face of a misspecified simulator. I am very short on time for ICML reviews. Apologies for my reviews being a bit short.

**Claims And Evidence:**

The authors claim that using a small labeled real dataset can help calibrate posterior inference from a misspecified simulator. The experiments are extensive and indeed showcase this behavior.

**Essential References Not Discussed:**

The authors may want to additionally cite two recent works to increasing robustness for SBI, which both work with *unlabeled* real datasets: https://arxiv.org/abs/2501.13483, https://arxiv.org/abs/2502.04949

They may also check if there are more relevant references about misspecification in SBI in these papers worth citing here as well.

**Experimental Designs Or Analyses:**

The experiments and their evaluation looks sensibel to me.

**Methods And Evaluation Criteria:**

The examples make sense indeed. As previously I continue to think that *labeled* real datasets (i.e. real datasets where the true parameters are known), is a rather niche situation. Yet I do acknowledge that there are relevant applications, where we do indeed have such a dataset.

**Other Comments Or Suggestions:**

none

**Other Strengths And Weaknesses:**

see above

**Questions For Authors:**

The authors have previously answered all my questions already during out submission processes.

**Relation To Broader Scientific Literature:**

Model misspecification is one of the major issues in SBI. The authors make an important contribution to this literature.

**Theoretical Claims:**

No proofs in the paper.

---

> ### Author Rebuttal · Authors · 2025-03-31
>
> We sincerely appreciate your review and your recognition of our contribution to the literature on simulation-based inference (SBI) under model misspecification.
>
> Regarding your concern about the availability of labeled real datasets, we acknowledge that such datasets are not always accessible. However, as discussed with Reviewer x5KJ, moving away from labeled calibration data in method papers introduces the challenge of defining alternative evaluation metrics. Without access to true parameters, validating the reliability of a method becomes significantly more difficult. That said, we agree that exploring approaches leveraging unlabeled real datasets is an interesting direction for future work.
>
> We will incorporate the suggested references on increasing robustness in SBI and check whether they contain additional relevant citations on misspecification.
>
> Thank you again for your thoughtful feedback and your time.

---

### Official Review · Reviewer_x5KJ · 2025-03-06

**Overall Recommendation:** 4

**Summary:**

This paper introduces Robust Posterior Estimation (RoPE), a method for addressing model
misspecification in simulation-based inference (SBI). Standard SBI algorithms often
assume a well-specified simulator, leading to biased posterior approximations when this
assumption is violated. RoPE mitigates this problem in scenarios where a small
calibration set of real-world observations paired with ground-truth parameters is
available. The method first learns a neural embedding via a standard SBI approach
(neural posterior estimation, NPE). It then fine-tunes this embedding using the
calibration set. Crucially, RoPE solves an optimal transport (OT) problem between the
embedded simulated data and the embedded observed data from the calibration set. At
inference time, the OT solution is used to obtain the posterior as a weighted sum of
posteriors conditioned on simulated data. The paper evaluates RoPE on six benchmark
tasks (two existing, four new) and reports performance compared to baselines.

## Update after rebuttal
I thank the authors for their detailed rebuttal, which successfully addressed all my questions and concerns. I agree with their proposed handling of the formal SBI baseline description (referencing the appendix). My initial positive evaluation stands, and I believe the paper makes a valuable contribution to the SBI literature.

**Claims And Evidence:**

The paper's claims center around RoPE's ability to provide accurate posterior inference
in the presence of model misspecification. These claims must be understood within a
specific, and somewhat less common, context within the broader field of simulation-based
inference (SBI): the availability of a small calibration set consisting of real-world
observations paired with corresponding ground-truth parameter values. Many typical SBI
applications operate without access to such paired data, even for a small calibration
set. While the paper acknowledges this setting (citing examples like cardiovascular
system inference and soft sensor development), a more thorough discussion of the
limitations this assumption places on RoPE's general applicability would strengthen the
work. The claims are therefore best interpreted as statements about performance
conditional on the availability of this calibration data. The authors provide new
benchmark task, which is positive, but validation with real world applications would
have been a stronger support.

Beyond the contextual considerations, the claims are generally supported by a
combination of theoretical justifications (based on an adapted definition of model
misspecification for SBI) and empirical evaluations across several benchmark tasks. This
is a minor point, but the clarity of the evidence presentation, particularly in Figure
1, somewhat hinders a complete and immediate assessment of these claims across all
tested scenarios. The dense and complex nature of Figure 1 makes it challenging to
definitively confirm RoPE's superior performance against all baselines on each specific
task without significant effort. This difficulty in visually parsing the results
slightly reduces the "convincing" aspect of the evidence, although the underlying data
may well support the claims.

**Essential References Not Discussed:**

No, the adapted definition model misspecification is stated clearly in the introduction
and the related work on this topic is discussed in the corresponding section.

**Ethical Review Flag:**

Flag this paper for an ethics review.

**Experimental Designs Or Analyses:**

I reviewed the experimental design. The choice of baselines is generally appropriate.
However, as mentioned above, Figure 1 is overloaded and makes it difficult to assess the
results clearly. A more detailed explanation of the "SBI" baseline in the main text is
needed, as are explanations of the MLP's strong performance (despite being based on
Gaussian posteriors), and of the performance of RoPE for different tasks.

**Methods And Evaluation Criteria:**

The introduction of four new benchmark tasks is a valuable contribution to the SBI
literature, providing standardized testbeds for evaluating methods under model
misspecification. However, the chosen evaluation metrics, LPP and ACAUC, while relevant,
are less common in the broader SBI literature.  Using more established metrics could
facilitate comparisons with other SBI methods and improve the interpretability of the
results for a wider audience.  Specifically, the negative log probability of the true
test parameter (NLTP), as described in Lueckmann et al. (2021) (and used e.g.,
Papmakarios et al. 2016, 2017, Greenberg et al. 2019), would provide a direct measure of
posterior accuracy (and is closely related to the LPP used here, but more standard).
Furthermore, the ACAUC metric, as implemented, assesses calibration on a per-parameter
basis. This approach might miss joint miscalibrations across multiple parameter
dimensions.  Considering metrics like expected coverage (as used in, e.g., Miller et
al., Deistler et al., Hermans et al.) could offer a more comprehensive assessment of
calibration, capturing potential dependencies in the posterior uncertainty. While LPP
and ACAUC provide some insight, supplementing them with (or replacing them by) NLTP and
expected coverage would strengthen the evaluation.

**Other Comments Or Suggestions:**

No.

**Other Strengths And Weaknesses:**

Originality: The paper presents a novel approach (RoPE) using optimal transport. The
introduction of new benchmark tasks also contributes to originality.

Significance: Model misspecification is a critical challenge in SBI. RoPE offers a
potential solution, but its significance is somewhat limited by the reliance on a
calibration set, which may not be available in all SBI applications.

Clarity:  The paper is well-written and well-structured overall. However, the
presentation could be significantly improved in terms of figures. Figure 1 is extremely
difficult to read and interpret. The lack of a clear concept figure (as shown in
Appendix B) makes it harder to grasp the overall approach at the beginning. Explanations
for certain results (SBI baseline, MLP baseline, RoPE performance on different tasks)
are currently insufficient.

**Questions For Authors:**

1. Figure 1 Overload: Figure 1 is extremely difficult to interpret due to the sheer
   number of lines, markers, and overlapping information. Could you simplify this
   figure, perhaps by splitting it into multiple figures, or by using a different
   visualization strategy? Also, please explicitly state in the caption what constitutes
   a "good" value for LPP and ACAUC (e.g., "Higher LPP values indicate better
   performance; ACAUC values closer to zero indicate better calibration").
2. "SBI" Baseline Explanation: The "SBI" baseline is not clearly explained in the main
   text. Please provide a more detailed description of how this baseline is trained and
   evaluated. Specifically, clarify what data is used for training and testing this
   baseline.
3. MLP Baseline Performance: Why does the MLP baseline perform so well, even though it
   assumes a Gaussian posterior? Are the true posteriors in the benchmark tasks close to
   Gaussian, or is there another explanation?
4. RoPE performance variations: It is unclear why the performance of RoPE varies across
   the different tasks. For example, explain in more details the performance for the SIR
   example. It was stated that the SIR task might be less misspecified and that NPE thus
   performs better. However, does this imply that for well-specified use-cases, RoPE
   will perform worse in general, e.g., be overly underconfident like for the SIR
   example? E.g., how does RoPE perform on the standard benchmarking tasks defined in
   Lueckmann et al. 2021 that are readily available in the `sbibm` package?
5. Calibration Set Assumption: The paper relies on the availability of a calibration set
   with ground-truth parameter values. While this is a valid scenario, it's less common
   in many SBI applications. Could you discuss the limitations of RoPE in settings where
   such a calibration set is not available? Are there potential extensions or
   modifications to RoPE that could address such scenarios?

**Relation To Broader Scientific Literature:**

The paper builds upon the SBI literature and addresses model misspecification. It cites
key works and acknowledges the unreliability of SBI under misspecification. The use of
optimal transport is a novel contribution in this specific context. The paper
differentiates itself by focusing on the scenario with a small, labeled calibration set.

**Theoretical Claims:**

I did not thoroughly check all proofs in the supplementary material, focusing instead on
the conceptual soundness and experimental validation. The optimal transport formulation
appears standard.

---

> ### Author Rebuttal · Authors · 2025-03-31
>
> Thank you for the thoughtful and constructive feedback. We appreciate your insights and will use them to improve the final version of our manuscript. Below, we detail the modifications we plan to make in response to your comments.
> ## Figure 1 is overly dense
> We acknowledge that Figure 1 contains too much information and will revise it to improve readability. We will use the additional space to split the figure in two. This will allow us to make the individual plots larger and have space to add the method labels directly on the plots, making them easier to parse. We will also implement the suggested changes.
> ## Clarification on NLTP and LPP
> We confirm that NLTP and LPP are exactly the same metric. We will make this explicit in the text to avoid confusion.
> ## ACAUC and joint calibration
> ACAUC quantifies marginal calibration by measuring the gap between empirical and expected coverage, averaged over confidence levels. However, it does not capture joint calibration, as dependencies between parameters are not explicitly assessed. Alternative dependence-sensitive metrics may require larger test sets to be stable. Given this limitation, we will explicitly discuss the trade-offs of ACAUC in the revised manuscript.
> ## SBI baseline details
> There is a formal description of the SBI baseline in the appendix (lines 965-970). Do you think it should be moved to the main text? The SBI baseline is trained and evaluated purely on simulated data. Since RoPE assumes a conditional independence structure, the SBI baseline should provide an upper bound on RoPE’s performance in terms of LPP.
> ## Performance of the MLP baseline
> The strong performance of the MLP baseline is likely due to the unimodal nature of the benchmark posteriors, where a Gaussian is a reasonable approximation. However, this baseline only performs well when given the largest calibration budget. For smaller budgets (200 and below), RoPE consistently outperforms it. Extending benchmarking tasks for SBI under misspecification with multi-modal posterior would be an interesting direction for future work.
> ## Performance variations of RoPE
> RoPE consistently returns calibrated and informative posteriors, as indicated by its LPP scores being higher than those of the prior. The results in Figure 1 are based on a fixed entropy regularization parameter, but in practice, this parameter would be tuned. With the chosen setting, RoPE can sometimes be slightly underconfident while remaining well-calibrated across all tasks. For Task E, using a smaller regularization value ($\gamma = 0.1$ or lower) resulted in stronger performance than the one we reported in the manuscript (with $\gamma=0.5$). We expect RoPE to match SBI closely on standard benchmarks when $gamma$ is small. Investigating further the effects of $\gamma$, parameter dimensionality, and test set size further would be interesting but probably goes beyond the scope of this work. Finally, we emphasize that when a simulator is well-specified, standard SBI methods should be preferred over RoPE, as RoPE is specifically designed for misspecified settings.
>
> ## The need for a labeled calibration set
> RoPE relies on a calibration set with ground-truth parameters, but as shown in our experiments, it can also be used without the fine-tuning step. In most benchmarks, this "OT-only" version already produces well-calibrated posteriors. However, the labeled test set was crucial in identifying that OT-only was insufficient for Task E. In a misspecified setting, learning a new model is necessary, and empirical data is required to validate its reliability. This justifies our assumption that a small set of real observations could be used for training. We nevertheless agree with you that labeled calibration data is not always available in real-world applications. In such cases, practitioners must define metrics that assess whether OT-only provides sufficient performance. In some applications, a fine-tuning step similar to RoPE’s could still be justified without labeled data, but this would require application-specific validation strategies. We will clarify these points in the final version of the paper.

---

> > ### Comment · Reviewer_x5KJ · 2025-04-08
> >
> > I thank the authors for their detailed rebuttal. All my concerns and questions have been addressed. Regarding the formal description of the SBI baseline I think it is fine keeping a pointer to the appendix for a detailed description.
> >
> > Overall, I believe this paper will make a valuable contribution to the SBI literature, which is consistent with my initial positive evaluation. I look forward to seeing the final version.

---

### Official Review · Reviewer_vbi2 · 2025-03-15

**Overall Recommendation:** 4

**Summary:**

Because it is challenging to use simulation-based inference (SBI) under model mis-specification, this paper proposes optimal transport to model the distribution between SBI-simulated data and a set of observed data and then constructing posterior distributions with neural posterior estimation (NPE). It relies on a calibration set {(theta,x_0)} linking theta and x_0 to modify pi(theta|x_s) when observing x_0. The method is demonstrated on a number of simulated and real benchmark datasets.

**Claims And Evidence:**

Main Comment

p.2 col 1: the 3rd quantity that access is given to is "a small calibration set of labelled real world observations C composed of iid samples from p^*(theta,x_0)". I understand that this is required by the developed method. My question is whether it is actually likely in practice that the samples in the calibration set will really be a) samples from p^*, and b) iiid.

(theta,x_0) can quite credibly (and perhaps frequently) not be drawn from the representative population p^*. Following the medical example in para 2 of the introduction, collecting (theta,x_0) can be very expensive. The decision to take these measurements could then credibly be selectively made based on estimates of quantities that are direct functions of x_0 or theta. E.g. (i) we should only measure the most severe cases, or (ii) those cases that will produce decisions/data that will like on some decision boundary (operate or not). What won't happen is that a random, iid decision (coin flip) will be made whether to make the measurement for each presenting patient (which would represent draws from p^*).

My argument here is that the (theta,x_0) calibration set won't be iid draws from p^*, but will be draws from some distribution p^** (say) that is dominated by (has a smaller domain than) p^*. In which case, if you assume the calibration set are draws from p^*, when you actually observe data from a patient who lies comfortably outside the observed domain of p^*, your modelling will go awry.

So while I like the paper as written, I think it would be healthy to:

* Understand how one might practically handle this (p^** not p^*) within the context of the method; What extra information would you need, and could credibly expect to have available? Etc.

* Perhaps see (in a toy example only) what would happen with posterior estimation with the current method when you in fact have samples from p^** but you assume these are from p^* (i.e. something will go wrong), and how correcting for this might correct the analysis.

This will go a long way to making this method more applicable in practice.

Related Comment

My understanding of (3) (p. 3 col 2) is that you're assuming that the sufficient statistics of x_s for model p, are the same as the sufficient statistics of x_0 for p^*. Given that the premise of the paper is that p is a mis-specified version of p^*, how likely is this assumption? I appreciate that the following paragraph after (3) discusses this to some extent. However, I'd be interested in understanding more what happens within the method if p is increasingly "too simple" so that (3) becomes increasingly violated. At what point does the method break down or possibly perform poorly? (Discussion response in the text is adequate.)

**Essential References Not Discussed:**

Good, to the extent of my knowledge.

**Ethical Review Concerns:**

.

**Experimental Designs Or Analyses:**

Reviewed for intuitive sense.

**Methods And Evaluation Criteria:**

Ok.

**Other Comments Or Suggestions:**

Small comments:
* p.2 col 1. l.31 "which reduces uncertainty compared to the prior distribution" - perhaps nit-picking, but posteriors don't always have 'reduced uncertainty' (smaller spread) over the prior, especially in situations of prior mis-specification (for example).

* p.2 col 1. l.33-34: "the ... simulator S that ... approximates p^*". - again, perhaps nit-picking, but simulators are forward-simulation schemes. There may be a density function associated with them, but this density function (that approximates p^*) is not the simulator itself.

* p.2 col 2 text after eq (1): calling p(theta|x_s) the "true" posterior confused me for a while, as p^*(theta|x) is the true posterior, and p(theta|x) is just the model you're attempting to fit. Maybe adjust the text here?

Trivial typos:

* p. 1 col 1 l.-3 "absence of A paired datasetS of" clash between "a" and a plural.

* p.2 col 1 l.25 integral in p^*(x_0) should be d\theta not dx_0

* p.2 col 1 l.38 p^*(theta,x_0) not defined (the reader has to infer this).

* p.7 col 1 l.-12 "we see of J-NPE and" - delete "of"

* p.7 col 2 l. 12 refers to Figure 3. The paper only appears to have 2 Figures.

**Other Strengths And Weaknesses:**

See above & below comments.

**Questions For Authors:**

See above.

**Relation To Broader Scientific Literature:**

Good.

**Theoretical Claims:**

The contribution is essentially methodological, with no new theory.

---

> ### Author Rebuttal · Authors · 2025-03-31
>
> We sincerely thank the reviewer for carefully reading our paper and appreciate the very constructive feedback regarding the plausibility of finding a calibration set sampled from $p^\star$ in practical settings. We now discuss this question and other comments in detail.
>
> ## Source distribution of the calibration set
>
> We agree with the reviewer that in some settings, assuming the calibration set consists of samples from the distribution $p^\star(\theta, x_o)$ may be unrealistic. Nevertheless, in important practical scenarios, it may sometimes be possible to influence the calibration set collection process to ensure it is representative of the test case scenario. In the following discussion, we denote an “imperfect” calibration set distribution as $p^c(\theta, x_o)$.
>
> ### Case 1: $p^c(\theta, x_o)$ and $p^*(\theta, x_o)$ have the same support
> In this case, as the size of the calibration set increases, the fine-tuning step should still be able to correct the estimated sufficient statistic for all observations in the test set. If the calibration set is large, one could even perform importance sampling to ensure that the fine-tuning step optimizes the expected loss on the ideal calibration set distribution. However, for smaller calibration sets, the sensitivity of RoPE depends on whether the fine-tuned neural statistic estimator (NSE) can generalize to unseen observations. This is highly dependent on the application, particularly regarding the non-linearity of the model misspecification and the inductive bias of the NSE. If the NSE struggles to generalize, we expect the entropy-regularized optimal transport (OT) step can bring some additional robustness. Specifically, observations where the fine-tuned NPE performs well will naturally be matched to the appropriate simulated statistics, leading to accurate posterior estimates.  In contrast, for observations where the sufficient statistic does not generalize well, their representation in the embedding space may become unstructured. As a result, applying OT to these embeddings may lead to random matches with the simulated statistics, causing the posterior to revert to the prior distribution, filtered by the parameter space where the NPE is reliable.
>
> ### Case 2: $p^c(\theta, x_o)$ and $p^*(\theta, x_o)$ have disjoint support
> In this more extreme case, even an arbitrarily large calibration set may fail to provide RoPE with relevant training examples for test observations. Consequently, the fine-tuning step relies entirely on out-of-distribution generalization, which is highly problem-dependent. If the fine-tuned NPE cannot generalize at all to the test set, the fine-tuning process may be counterproductive. However, even in this situation, we expect the OT step to highlight this issue. Since real and synthetic observations will not match meaningfully, the transport matrix should approach a uniform and the posterior will revert to the prior.
>
> While this second scenario is particularly challenging to study, we propose to approximate "Case 1" in a controlled experiment on the Light Tunnel task. Specifically, we will construct a calibration set using samples from Prior B (as in Figure 2) and evaluate performance on a test set drawn from Prior A.
>
> If the paper is accepted, we plan to include this experiment in the appendix and add a discussion in Section 6 to highlight RoPE’s sensitivity to non-representative calibration sets, following the considerations outlined above.
>
> ## On the failure of RoPE when the simulator is increasingly simple
> We agree that further investigating the trade-offs between model complexity and robustness to misspecification is an important research direction but goes beyond the scope of our paper. In many applications, there exists a trade-off:
> - **Simplistic simulators** may ignore complex relationships between parameters and observations but have structural properties that generalize well to real-world data.  SBI models trained on these simulators may be less sensitive
> - **Highly complex simulators** may capture intricate relationships but risk overfitting to simulation-specific artifacts, failing to generalize under slight misspecifications.
>
> For RoPE, because it assumes conditional independence $x_o \perp \theta \mid x_s$, it is better suited for handling the latter case rather than highly simplistic simulators that fail to encode meaningful parameter-observation relationships. The conditional independence assumption acts as a backstop to shortcut learning, but for the same reason, RoPE’s posterior informativeness is inherently limited by the informativeness of the posterior learned from simulations, as transparently discussed in Section 3, Paragraph 2.
> ## Minor comments & typos
> We appreciate the reviewer’s attention to detail in pointing out minor phrasing inconsistencies and imprecisions. We will carefully edit the manuscript to address these comments.

---

> > ### Comment · Reviewer_vbi2 · 2025-04-02
> >
> > Thank you for your considered response, and for the discussion and responses of the other reviewers -- all very informative.
> >
> > With apologies to the authors, but I am in two minds about how/whether to adjust my score.
> >
> > * If the authors can contribute the described new analysis/discussion on the source distribution of the calibration set (great!), then I will be happy to increase my score 3->4.
> >
> > * However the paper does not currently have this analysis, and we simply have a promise that this will be included. (No slight against the word of the authors is intended here.) So on what is currently available in the non-revised manuscript, my score should remain at 3.
> >
> > Clearly my complaint (which has nothing to do with the authors of this paper) is that a peer review process that requires reviewers to make accept/reject recommendations on paper acceptance based on *promises* to do things has always seemed rather suspect. Many (other) author teams spend the few days between the release of reviews and the due date of rebuttals in an extremely unhealthy crunch to perform additional requested analyses. While I am glad that the current authors prioritise their health, such authors are implicitly penalised compared to the crunch teams due to the uncertainty that they will deliver on their promises. Is this (both unhealthiness and implicit penalisation) really something that the ML community really wants to promote?
> >
> > Apologies again (particularly to the authors) for the rant. I will increase my score to 4 and trust the authors, as the only fair outcome for this nice paper. Obviously I need to go and lie down now.

---

> > > ### Author Response · Authors · 2025-04-08
> > >
> > > Dear Reviewer,
> > >
> > > Thank you for trusting our good faith and raising your score to 4. We're happy to share preliminary results from the experiments we committed to including in the final version of the paper. You can find them anonymized [here](https://anonymous.4open.science/r/RoPE_ICML_Rebuttal-4246/README.md).
> > >
> > > While these results currently lack error bars, they already support the discussion presented in our rebuttal and which we will include in Section 6. Specifically, they show a performance drop for RoPE but not a complete collapse as RoPE keeps improving upon the prior while being calibrated (at all calibration set size above 10). They also highlight that, in this situation, RoPE performs well on samples similar to those seen in the calibration set, but struggles with out-of-distribution samples. While this is only a partial eye on the issue you mentioned, whose impact will strongly depend on the exact setup considered, we believe these empirical results are consistent with the theoretical discussion we will add to section 6.
> > >
> > > We believe this additional experiment will provide valuable insights to readers, and we thank you again for raising this important point.
> > >
> > > With our best regards,
> > > The authors

---

### Official Review · Reviewer_PVCR · 2025-03-17

**Overall Recommendation:** 4

**Summary:**

This paper presents a method for improving simulation-based inference when the simulator is misspecified. It combines neural posterior estimation with optimal transport, using a potentially small labeled calibration real data paired to corresponding parameters to correct for misspecified simulator. It then fine-tunes a representation using the calibration data and uses OT to align the distributions of representations from simulated and real data.

**Claims And Evidence:**

The main claim from the paper is that the method RoPE can provide calibrated posterior distributions under simulator misspecification, even with a small calibration set. There are a few experiments with synthetic data ,and two real-world benchmark tasks (Figure 1), showing improved log-posterior probability and Average Coverage AUC compared to baselines and methods. Ablation studies demonstrating the importance of both the fine-tuning and OT steps. This is a well supported evidence to me.

There is also a claim that RoPE is robust to prior misspecification. Experiments on extensions of tasks C and E (figure 2b, 2c and appendix C), where the true parameter distribution differs from the prior -limited to a single misspecification, showing that RoPE maintains performance. Somewhat the least convincing claim given the limited scope of the priors selections.

Another claim is that RoPE offers a controllable balance between calibration and informativeness via hyperparams. Figure 2a shows an effect of gamma hyperparam on posterior sharpness and calibration. Sounds good to me.

Finally, a claim that adding a calibration set in the way specified can improve on standard SBI. This is overall well discussed and non-ambiguous to me.

**Essential References Not Discussed:**

My understanding of RoPE also connects to many other areas of machine learning which I feel the paper is missing:
* semi-supervised learning: fundamentally RoPE can be framed as a semi-supervised method, but this is not explicitly discussed.
* domain adaptation: RoPE can also be viewed as a DA method, with the simulator as the source domain and the real world as the target domain. There are a set of domain adaptation papers which attempt to bridge the gap between theory and data. sim2real (in robotics ) is also a very relevant field.

**Experimental Designs Or Analyses:**

Synthetic data (tasks A, B, C, D): generally well-designed to test different aspects of misspecification. The use of established benchmarks (CS, SIR) are all good to me. Tasks E, F are beyond purely synthetic data but simple lab setups, not really real-world applications.
Baselines are all well and confined within the definitions of the paper.

Metrics: LPP and ACAUC are reasonable choices for evaluating posterior quality. The discussion of their limitations is helpful.

I have nothing much to critique on the experiments. They are all well done within its limited scope anyway: the experimental designs are limited by the small size of the calibration set, small number of parameters, and the conditional independence assumption.

**Methods And Evaluation Criteria:**

The methods and evaluations are internally consistent and appropriate given the assumptions and limitations. The bigger question is whether those assumptions and limitations are too restrictive for broad real-world applicability.

RoPE combines existing techniques (NPE, OT) in a novel way on a hard problem. The method is clearly described, with equations and algorithm. The core modeling assumption is explicitly stated (eqn 3), and the fine-tuning objective (eqn 6) is also well defined.

The evaluations use a mix of synthetic and real-world benchmarks, simplified lab setups, and comparisons to many baselines and competing methods. In fact there is a wealth of evaluations, always within the scope. Metrics are appropriate for evaluating posterior calibration and informativeness.
Ablation studies OK to me too.

**Other Comments Or Suggestions:**

* OTT is not defined (had to look the reference).
I feel the authors could also propose a setting where RoPE could be useful: design an experiment with a small calibration set, and one may be good to go with SBI. This is perhaps not such a strong requirements in some experimental sciences, while almost impossible in most observational sciences.

**Other Strengths And Weaknesses:**

I mentioned a few strengths and weaknesses, here are some others:
**Strengths**:
* one of the very few paper dealing with SBI misspecification which is one of the main SBI limitation
* intuitive combination of NPE and OT for handling misspecification
* well-written paper,  clear problem formulation and motivation.
* scoped experiments and evals
* open about limitations (this is a hard problem)

**Weaknesses**: (most already mentioned)
* limited setting: requirement of a labeled calibration set and the focus on low-dimensional parameters (also a limitation of SBI though). Similarly this is not clear how it would scale to larger and more diverse datasets, especially if calibration set does not grow simultaneously
* dependence on conditional independence: limiting.

**Questions For Authors:**

Not too many questions, beyond extending to other non-SBI simu-real gaps literature.

**Relation To Broader Scientific Literature:**

The paper cites adequately SBI, OT and misspecification (in the Bayesian sense) literature.

**Theoretical Claims:**

The theoretical foundation, particularly its self-calibration property and the validity of its posterior factorization  relies on the conditional independence assumption eqn (3). This assumption is acknowledged by the authors, and necessary but represents a significant limitation. It basically says that the simulator completely mediates the relationship between parameters and real-world observations. Thus the theoretical guarantees and practical performance can degrade when it breaks,  and leads to wrong inferences.

---

> ### Author Rebuttal · Authors · 2025-03-31
>
> We sincerely thank the reviewer for their thoughtful and constructive feedback. We greatly appreciate the careful assessment of our work and the valuable insights provided. Below, we address the key points raised.
> ## Additional References
> We completely agree that RoPE is closely related to semi-supervised learning and Sim2Real literature. If accepted, we will use part of the additional page to elaborate on these connections.
> ## Setting Limitations
> As demonstrated in our experiments, in the absence of a calibration set, the version of RoPE without fine-tuning (OT-only) provides a viable way to handle model misspecification in some benchmarks. Nevertheless, as noted in the introduction, when posterior estimates need to be reliable, a labeled dataset is crucial for empirical validation and it appears thus natural to save a subset of this dataset for training purposes. We also recognize that settings where no labeled data is available exist. However, evaluating methods in such scenarios is highly application-dependent as it requires appropriate unsupervised metrics and we believe it may be difficult to create one generic method to handle these diverse set of problems. While we hope that RoPE’s fine-tuning step can be adapted to leverage unsupervised data and metrics specific to these applications, we acknowledge that this remains an open question. We believe future work focused on adapting RoPE to these real-world use cases could be highly impactful.
>
> Regarding scalability, as acknowledged in Section 5, we have not yet explored RoPE’s performance in medium- or high-dimensional parameter spaces, and its ability to correct for misspecification in those settings remains an open challenge. As the reviewer pointed out, model misspecification in SBI is a complex issue, and it is unlikely that any single method can address all possible cases.
> ## Conditional Independence
> We appreciate the reviewer’s careful consideration of RoPE’s assumptions. As noted in Section 3, the independence assumption is indeed a strong constraint. It limits RoPE’s ability to uncover dependencies between observations and parameters that are not explicitly captured by the simulator. If this assumption does not hold, RoPE cannot recover the true posterior, even with an arbitrarily large calibration set. However, RoPE will approximate the posterior by leveraging the simulator’s known dependencies, which may contribute to robustness and interpretability. Furthermore, please note that this conditional independence assumption does not hold for the two real-data testbeds (tasks E and F), but RoPE still returns informative posterior estimates.

---

> > ### Comment · Reviewer_PVCR · 2025-04-09
> >
> > Thank you for your answers. My comments have been addressed. I am looking forward for the revised version.

---

### Decision · Program_Chairs · 2025-05-01

**Decision:**

Accept (oral)

**Comment:**

This paper presents an approach for simulation-based inference when the simulator is misspecified. This is an important and timely topic, especially considering the ubiquitousness of misspecification. The approach combines neural posterior estimation with optimal transport, using a set of observation/simulation pairs for calibration. Sufficient statistics are learnt within the approach and calibration data is used to align resulting distributions from simulated and real data by OT. The paper was reviewed by a team of four who all recommended acceptance. The paper was found to be well-written, presenting a wealth of convincingly esposed and analysed test cases, and open about potential/current limitations of the proposed approach. I would like to congratulate the authors on this promising work and add a few questions below that I hope could be useful towards an improved version.

page 2: is the dimension l (lower case L) related to k?

p_phi: a word could be said about the class

Equation (1): the argmax may be a set...

Perfect surrogate: potentially misleading terminology

For all x_o before (Cannon et al. 2022): why not quantify the variable just before p(x_o|theta)?

Page 3, about “there exists S”: so, this S can potentially be a single couple in the product space, right?

Besides this, same question about the logical quantifier arriving at the end. Written in full words at the end of the sentence (for all…) it would appear absolutely alright, but with the logical quantifier it feels awkward to me.  This is more about polishing, but I would recommend checking best practices on that.

At the bottom of the page: better use parentheses in rho/(rho + gamma). So in case gamma=0, tau=1 would apply for any rho>0?

Page 4, “In our setting, an ideal coupling would pair each real-world observation with the simulation generated by the same parameter”: this may sound like simulations are deterministic. Is it more in the sense that there is only one simulation per observation in the pairs used for calibration?

Just below, regarding h_o: it seems unclear at this stage where h_o is coming from.

Results: it would be worth looking in more detail at what happens with Task B and why the approach does not adapt well there.

References: the same paper Delaunoy et al. is cited twice with two different dates. Please check that references are in order and when preprint versions have advanced to publication, cite the resulting publication.